# Architecture-Aware Generalization Bounds for Temporal Networks: Theory and Fair Comparison Methodology

## Abstract

Learning from time series is fundamentally different from learning from i.i.d. data: temporal dependence can make long sequences effectively information-poor, yet standard evaluation protocols conflate sequence length with statistical information. We propose a dependence-aware evaluation methodology that controls for effective sample size $N_{\text{eff}}$ rather than raw length $N$, and provide end-to-end generalization guarantees for Temporal Convolutional Networks (TCNs) on $\beta$-mixing sequences. Our analysis combines a blocking/coupling reduction that extracts $B = \Theta(N/\log N)$ approximately independent anchors with an architecture-aware Rademacher bound for $\ell_{2,1}$-norm-controlled convolutional networks, yielding $O(\sqrt{D \log p / B})$ complexity scaling in depth $D$ and kernel size $p$. Empirically, we find that stronger temporal dependence can *reduce* generalization gaps when comparisons control for $N_{\text{eff}}$ - a conclusion that reverses under standard fixed-$N$ evaluation, with observed rates of $N_{\text{eff}}^{-0.9}$ to $N_{\text{eff}}^{-1.2}$ substantially faster than the worst-case $O(N^{-1/2})$ mixing-based prediction. Our results suggest that dependence-aware evaluation should become standard practice in temporal deep learning benchmarks.

## 1 Introduction

Modern deep architectures, notably Temporal Convolutional Networks (TCNs) Lea et al. (2017); Bai (2018) and Transformer variants Vaswani et al. (2017), underpin state-of-the-art forecasting and representation learning across domains ranging from clinical monitoring to large-scale operational forecasting and management Lim et al. (2021); Oreshkin et al. (2019). Despite this success, two fundamental gaps limit our understanding of temporal deep learning.

Gap 1: Evaluation on dependent data is confounded. A common practice is to compare models by varying the raw sequence length $N$ or by holding $N$ fixed while changing dependence strength (e.g., correlation). However, for dependent sequences, $N$ is a poor proxy for the amount of statistical information: strong temporal correlation can drastically reduce the number of effectively independent observations ("effective sample size")Geyer (1992); Sokal (1997). As a result, "standard" comparisons at equal $N$ conflate two distinct effects: (1) changes in temporal structure (dependence) and (2) changes in information content. This confounding can systematically bias conclusions about if dependence helps or hinders learning.

Gap 2: Architectural scaling under dependence lacks clear guarantees. Classical generalization analyses rely on independence and therefore do not directly apply to time series. While mixing-based learning theory Yu (1994); Kuznetsov & Mohri (2014b) provides tools to analyze dependence, it often does not expose how modern architectural choices (depth, kernel size, norm control) affect sample complexity in deep temporal models. In contrast, norm-based i.i.d. analyses yield explicit architectural dependence (e.g., $\sqrt{D}$ rather than exponential in depth) under explicit norm control Neyshabur et al. (2015); Golowich et al. (2018). A central challenge is to retain such architecture-aware scaling while handling temporal dependence.

Our approach: effective-sample-size matching with supporting theory. We address these gaps with a methodology-first approach. On the empirical side, we introduce evaluation protocols that control for an **effective sample size** $N_{\text{eff}}$, i.e., a proxy for the number of "nearly independent" learning-relevant observations contained in a length-$N$ dependent sequence. Because $N_{\text{eff}}$ is not uniquely defined in general, we

adopt a definition that is *aligned with our theory*: under $\beta$-mixing, the key control quantity in our bounds is the *anchor count* induced by blocking (denoted $B$), and we instantiate $N_{\text{eff}}$ so that comparisons equalize this effective information budget. This enables comparisons across dependence regimes by separating changes in information content from changes in temporal structure. On the theoretical side, we combine a blocking/coupling reduction for $\beta$-mixing sequences with an i.i.d. architecture-dependent complexity bound for norm-controlled convolutional networks (via $\ell_{2,1}$ filter-group norm constraints). The resulting bounds are conservative but provide a baseline: they establish learnability under dependence and make explicit how architectural scaling laws interact with effective information.

We make three contributions:

1. **Fair-comparison methodology for dependent sequences.** We propose to match $N_{\text{eff}}$ rather than raw $N$ when the goal is to compare models or dependence regimes on equal information budgets.

2. **Empirical findings enabled by fair comparison.** Applying this methodology to synthetic autoregressive processes and physiological sequences reveals regimes in which stronger dependence is associated with smaller generalization gaps at fixed $N_{\text{eff}}$, a phenomenon that is obscured (and can appear reversed) under fixed-$N$ evaluation.

3. **Architecture-aware generalization baseline under $\beta$-mixing.** We provide end-to-end bounds for TCNs on exponentially $\beta$-mixing sequences, achieving explicit dependence on depth (via a $\sqrt{D}$ factor) and mild polylogarithmic dependence on kernel size. Under exponential $\beta$-mixing, the dependent-to-i.i.d. reduction yields an effective anchor sample size $B = \Theta(N/\log N)$, inducing an additional $\sqrt{\log N}$ factor relative to the i.i.d. $1/\sqrt{N}$ rate.

We distinguish between **standard evaluation** (comparisons at fixed raw length $N$) and **fair comparison** (comparisons that control for effective sample size $N_{\text{eff}}$).

Section 2 reviews related work on dependent-data learning and norm-based complexity control. Section 3 provides preliminaries. Section 4 presents our dependence-aware generalization baseline, and Section 5 reports empirical results. Section 6 presents a discussion. We conclude in Section 7; the appendix contains full proofs and additional experimental details.

## 2   Related Work

**Generalization under dependence.** Classical PAC-style generalization theory is typically developed for i.i.d. samples, while time series violate this assumption. A long line of work studies concentration and uniform convergence for stationary mixing processes, often via blocking/coupling arguments that reduce dependent sequences to collections of approximately independent blocks (e.g., early empirical-process rates for $\beta$-mixing sequences Yu (1994), nonparametric time-series prediction through adaptive model selection Meir (2000), and surveys of mixing tools Bradley (2005)). Building on these ideas, Mohri and Rostamizadeh Mohri & Rostamizadeh (2008) developed Rademacher-complexity bounds for $\beta$-mixing sequences, and later stability-based bounds were also derived for mixing processes Mohri & Rostamizadeh (2009). Alternative dependent-learning viewpoints include discrepancy-based generalization analyses Kuznetsov & Mohri (2014b), as well as PAC-Bayes approaches for weakly dependent sequences (e.g., Alquier & Guedj (2018)). Recent work by Abélès et al. Abeles et al. (2024) proposes an online-to-PAC framework with delayed feedback to control dependence, which is complementary to our focus here. We focus on absolute regularity ($\beta$-mixing) because it supports total-variation coupling, the key tool behind our anchor-based dependent-to-i.i.d. reduction. Alternative online-learning approaches (e.g., sequential Rademacher complexity Rakhlin et al. (2010)) handle adversarial sequences but express guarantees in regret terms rather than sample complexity.

**Norm-based generalization for deep networks (i.i.d.).** Classical VC-dimension analyses provide architecture-dependent bounds that scale with parameter counts Bartlett et al. (1998; 2019), but these can be loose for modern over-parameterized networks. Norm control has become a standard way to obtain architecture-dependent generalization guarantees for deep networks. For feedforward networks, seminal bounds scale with products of layer norms and improve depth dependence relative to VC-style analyses

(e.g., Neyshabur et al. (2015); Bartlett et al. (2017); Golowich et al. (2018)). These results motivate using norm budgets as a proxy for function-class capacity and help explain why deep networks can generalize despite over-parameterization. Our theoretical ingredient follows this tradition: we rely on a norm-controlled complexity bound for convolutional layers and then lift it to dependent data through a blocking/coupling reduction.

**Convolutional networks and weight sharing.** Generalization analyses for CNNs must explicitly account for parameter sharing and the structured linear operators induced by convolution. Long and Sedghi Long & Sedghi (2019) provide generalization bounds for deep CNNs that are independent of the input resolution/feature-map size, highlighting the role of shared parameters. Ledent et al. Ledent et al. (2021) develop norm-based bounds for deep multi-class CNNs and incorporate weight sharing directly into the Rademacher/covering analysis. More recent refinements study structure and filter-level norms, yielding potentially tighter bounds for CNN-like architectures Galanti et al. (2023), and related capacity-measure investigations examine how much "excess capacity" standard norm bounds may permit in modern architectures Graf et al. (2022). These CNN results serve as the appropriate i.i.d. architectural baseline for our temporal convolutional setting.

**Temporal deep models: TCNs and Transformers.** Temporal Convolutional Networks (TCNs) Lea et al. (2017); Bai (2018) are widely used for forecasting and sequence modeling due to causal/dilated convolutions and large receptive fields. Generalization analyses for recurrent or sequence models under dependence exist (e.g., mixing-based bounds for RNN-style predictors Kuznetsov & Mohri (2014a)), but they do not directly yield the simple architectural scaling laws we seek for TCNs. For Transformer-style models, several works analyze generalization through norm control or stability perspectives (e.g., sequence-length independent norm-based bounds Trauger & Tewari (2024) and algorithmic viewpoints for in-context learning Zhang et al. (2024)). Our paper is centered on temporal convolutions and on the interaction between dependence and inductive bias under a controlled information budget, rather than on deriving architecture-specific bounds for attention.

**Evaluation methodology and effective sample size.** A practical but under-emphasized issue in empirical time-series ML is that comparing settings at fixed raw length $N$ can silently change the information budget when dependence strength changes. Effective sample size $N_{\text{eff}}$ is a classical way to quantify information loss due to correlation, it appears through variance-inflation/integrated-autocorrelation-time identities and is widely used in time-series statistics and MCMC diagnostics Geyer (1992); Sokal (1997). Motivated by this, our empirical protocol matches $N_{\text{eff}}$ across dependence regimes to isolate the effect of temporal structure from the effect of information content.

## 3 Preliminaries

To analyze generalization for temporal models trained on dependent data, we use three ingredients: (i) a dependence model for time series ($\beta$-mixing), (ii) a capacity measure for the hypothesis class (Rademacher complexity, used as an i.i.d. ingredient), and (iii) an information proxy that enables fair empirical comparisons (effective sample size $N_{\text{eff}}$). We develop these tools in a form tailored to window-based prediction with TCNs.

**Learning from a single dependent time series.** Let $\{z_t\}_{t=1}^N$ be a stationary time series with $z_t \in \mathbb{R}^n$. We consider one-step-ahead prediction using a fixed-length context window of length $q \geq 1$. Define supervised examples

$$x_t = (z_{t-q+1}, \dots, z_t) \in \mathbb{R}^{q \times n}, \tag{1}$$

$$y_t = z_{t+1} \in \mathbb{R}^n, \quad t = q, \dots, N-1. \tag{2}$$

yielding $m = N - q$ dependent examples $\{(x_t, y_t)\}_{t=q}^{N-1}$ from a single sequence.

Given a predictor $f : \mathbb{R}^{q \times n} \to \mathbb{R}^n$ and loss $\ell : \mathbb{R}^n \times \mathbb{R}^n \to \mathbb{R}_+$, the population risk and empirical risk are $\mathcal{L}(f) = \mathbb{E}[\ell(f(x_t), y_t)]$, $\widehat{\mathcal{L}}_m(f) = \frac{1}{m} \sum_{t=q}^{N-1} \ell(f(x_t), y_t)$, where the expectation is w.r.t. the stationary law of the process. Our goal is to control the generalization gap $|\mathcal{L}(f) - \widehat{\mathcal{L}}_m(f)|$ despite temporal dependence.

**Stationary $\beta$-mixing processes.** Stationarity ensures that the distribution of finite windows does not change over time. To quantify dependence, we use $\beta$-mixing. Let $U_t = (x_t, y_t)$ denote the example process and let $\mathcal{F}_a^b = \sigma(U_s : a \leq s \leq b)$. We also write $\mathcal{F}_{-\infty}^t = \sigma(U_s : s \leq t)$ and $\mathcal{F}_{t+k}^\infty = \sigma(U_s : s \geq t + k)$. Throughout, $\beta(\cdot)$ refers to the $\beta$-mixing coefficients of $\{U_t\}$. The $\beta$-mixing coefficient at lag $k$ is: $\beta(k) = \sup_{t \geq 1} \mathbb{E}\left[\sup_{A \in \mathcal{F}_{t+k}^\infty} \left|\mathbb{P}(A \mid \mathcal{F}_{-\infty}^t) - \mathbb{P}(A)\right|\right]$. A small $\beta(k)$ means observations separated by $k$ steps are nearly independent.

**Assumption 1** (Exponential $\beta$-mixing)**.** *There exist constants $C_0, c_0 > 0$ such that for all $k \geq 0$, $\beta(k) \leq C_0 e^{-c_0 k}$.*

This condition is sufficient to justify a blocking/coupling reduction, which is the core technical step of Section 4.

**Remark 1** (Mixing of windowed examples)**.** *Let $\beta_z(\cdot)$ denote the $\beta$-mixing coefficients of the raw process $\{z_t\}$. Since each example $U_t = (x_t, y_t)$ depends only on $(z_{t-q+1}, \ldots, z_{t+1})$, the example process $\{U_t\}$ is also $\beta$-mixing and satisfies, for all $k > q$, $\beta_U(k) \leq \beta_z(k - q)$. Throughout this paper, $\beta(\cdot)$ refers to the mixing coefficients of the windowed example process $\{U_t\}$, and the delay parameter $d$ counts steps in the example index (i.e., anchors $U_i$ and $U_j$ are separated by $|i - j|$ steps).*

***Implication for the delay parameter:*** *If the raw process satisfies $\beta_z(k) \leq C_0 e^{-c_0 k}$, then for the windowed process we have $\beta_U(k) \leq C_0 e^{-c_0(k-q)}$ when $k > q$. To ensure $\beta_U(d + 1) \leq 1/m$, we need $d \geq q + (\log m)/c_0$ rather than just $d \geq (\log m)/c_0$. Since $q$ is a fixed architectural hyperparameter (e.g., $q = 32$ in our experiments) and $\log m$ ranges from $\sim 6$ to $\sim 12$, the window size $q$ affects the constant in $B = \Theta(m/\log m)$ but not the asymptotic scaling. Concretely, with $q = 32$, $c_0 \approx 0.22$ (for $\rho = 0.8$), and $m \approx 18{,}000$, the effective delay is $d^* \approx 32 + 45 = 77$, yielding $B \approx 230$ anchors rather than the $B \approx 390$ that would result from ignoring $q$.*

**Rademacher complexity (i.i.d. ingredient)** Rademacher complexity quantifies the ability of a *real-valued* function class to fit random signs. For a class $\mathcal{F} \subseteq \{f : \mathcal{U} \to \mathbb{R}\}$ and an i.i.d. sample $S = \{u_i\}_{i=1}^M$, the empirical Rademacher complexity is $\widehat{\mathfrak{R}}_S(\mathcal{F}) = \frac{1}{M} \mathbb{E}_\sigma\left[\sup_{f \in \mathcal{F}} \sum_{i=1}^M \sigma_i f(u_i)\right]$, where $\sigma_i \in \{\pm 1\}$ are independent Rademacher variables. In i.i.d. learning, $\mathfrak{R}_M(\mathcal{F}) = \mathbb{E}_S[\widehat{\mathfrak{R}}_S(\mathcal{F})]$ controls generalization. In our analysis, we will apply this i.i.d. machinery to the real-valued loss class $\ell \circ \mathcal{F}_{D,p,R}$ after constructing an *approximately independent block sample* from the original time series (Section 4).

**Effective sample size and fair comparison.** For dependent data, the raw number of examples $m$ can be a misleading measure of information content. We therefore use an effective sample size $N_{\text{eff}}$, defined informally as the number of i.i.d. samples that would yield comparable concentration behavior. Our empirical protocol compares dependences at matched $N_{\text{eff}}$, not matched raw length.

**Relationship between theory and empirical calibration.** In our theoretical analysis (Section 4), the key quantity controlling generalization is the *anchor count* $B = \lfloor m/(d+1) \rfloor$ induced by blocking under $\beta$-mixing. In our empirical protocol, we use a classical ACF-based effective sample size $N_{\text{eff}}^{(\text{ACF})}$ (defined below) as a practical proxy for matching information budgets. These quantities are related but distinct: $B$ arises from the mixing-based reduction and scales as $\Theta(N/\log N)$ under our delay choice, while $N_{\text{eff}}^{(\text{ACF})}$ captures variance inflation due to autocorrelation. We use $N_{\text{eff}}^{(\text{ACF})}$ empirically because it is directly computable from $\rho$ and provides a principled way to match information content across dependence regimes, even though the theoretical bounds are stated in terms of $B$. Appendix A.1 elaborates on this distinction.

**AR(1) calibration (synthetic).** For a stationary AR(1) process with autocorrelation $\text{Corr}(z_t, z_{t+k}) = \rho^k$, the classical ACF-based approximation is $N_{\text{eff}}^{(\text{ACF})} \approx N \cdot \frac{1-\rho}{1+\rho}$, which follows from the integrated autocorrelation time $\tau_{\text{int}} = (1 + \rho)/(1 - \rho)$ Wilks (2011); Geyer (1992). To match information budgets across dependence strengths, we choose raw lengths via $N(\rho) = \lfloor N_{\text{eff}}^{(\text{ACF})} \cdot (1 + \rho)/(1 - \rho) \rfloor$. We use this calibration in later to isolate the effect of temporal structure from the effect of available information.

**Model class: causal TCNs and norm control.** We focus on causal TCN predictors built from 1D convolutions and ReLU activations, i.e., $\sigma(x) = \max(0, x)$ applied elementwise. ReLU is 1-Lipschitz and satisfies $\sigma(0) = 0$, properties used in the Rademacher complexity analysis of Section 4. Let a depth-$D$ TCN have con-

volutional weight tensors $W^{(\ell)} \in \mathbb{R}^{C_\ell \times C_{\ell-1} \times p}$, $\quad \ell = 1, \ldots, D$, with kernel size $p$ and channels $C_\ell$. To control capacity, we use a filter-group norm (an $\ell_{2,1}$-type norm over output filters): $\left\|W^{(\ell)}\right\|_{2,1} = \sum_{j=1}^{C_\ell} \left\|W^{(\ell)}_{j,:,:}\right\|_2$.

We impose the constraint $\|W^{(\ell)}\|_{2,1} \leq M^{(\ell)}$ for each layer $\ell$. We write $R = \prod_{\ell=1}^{D} M^{(\ell)}$ to denote the product of layer-wise norm budgets, which is the quantity that appears in norm-based, architecture-aware complexity bounds. **Architecture conventions:** The input $x_t \in \mathbb{R}^{q \times n}$ is treated as a sequence of $q$ time steps with $n$ features each; thus the channel dimension at layer 0 is $C_0 = n$. The kernel size $p$ is the temporal filter width: a 1D causal convolution with kernel size $p$ uses $p$ consecutive time steps (positions $i - p + 1, \ldots, i$) to compute the output at position $i$. In our experiments, $p = 3$. The hypothesis class of norm-controlled causal TCNs is then

$$\mathcal{F}_{D,p,R,B_f} = \Big\{ f_W : \mathbb{R}^{q \times n} \to \mathbb{R}^n \,\Big|\, \text{a depth-}D \text{ TCN}, \|W^{(\ell)}\|_{2,1} \leq M^{(\ell)} \,\forall \ell,$$
$$\prod_{\ell=1}^{D} M^{(\ell)} \leq R, \text{ and } \|f_W(x)\|_2 \leq B_f \,\forall \|x\|_F \leq B_x \Big\}. \tag{3}$$

The output bound $B_f$ is enforced via an output clipping layer $\text{clip}_{B_f}(y) = y \cdot \min(1, B_f/\|y\|_2)$. Since clipping is 1-Lipschitz, this does not increase Rademacher complexity: $\mathfrak{R}_B(\text{clip} \circ \mathcal{F}) \leq \mathfrak{R}_B(\mathcal{F})$. For notational convenience, we write $\mathcal{F}_{D,p,R}$ when $B_f$ is clear from context.

**Boundedness assumptions**

**Assumption 2** (Bounded inputs, outputs, and targets). *There exist constants $B_x, B_y, B_f > 0$ such that: (i) $\|x_t\|_F \leq B_x$ almost surely (equivalently $\|\text{vec}(x_t)\|_2 \leq B_x$), (ii) $\|y_t\|_2 \leq B_y$ almost surely. The output bound (iii) $\|f(x)\|_2 \leq B_f$ for all $\|x\|_F \leq B_x$ is enforced by the hypothesis class definition equation 3 via output clipping, not assumed on trained models.*

**Remark 2** (Dependence of $B_x$ on context window length). *Under Assumption 2, the input bound $B_x$ depends on the context window length $q$. If the raw observations satisfy $\|z_t\|_2 \leq b_z$ for all $t$, then $x_t = (z_{t-q+1}, \ldots, z_t) \in \mathbb{R}^{q \times n}$ satisfies $\|x_t\|_F \leq \sqrt{q} \cdot b_z$. Consequently, while weight sharing in TCNs prevents explicit scaling with sequence length in the Rademacher bound, the input bound $B_x$ introduces implicit dependence on $q$. In practice, $q$ is typically a fixed architectural hyperparameter (e.g., $q = 32$ in our experiments) rather than a quantity that grows with the total sequence length $N$, so this dependence does not affect our main scaling conclusions.*

**Assumption 3** (Lipschitz loss). *For each fixed $y$, the map $\hat{y} \mapsto \ell(\hat{y}, y)$ is $L$-Lipschitz in $\|\cdot\|_2$.*

For squared loss $\ell(\hat{y}, y) = \|\hat{y} - y\|_2^2$ and Assumption 2, $0 \leq \ell(\hat{y}, y) \leq (B_f + B_y)^2$, $\quad |\ell(\hat{y}, y) - \ell(\hat{y}', y)| \leq 2(B_f + B_y)\|\hat{y} - \hat{y}'\|_2$. Thus squared loss is $L$-Lipschitz with $L = 2(B_f + B_y)$. When applying results stated for losses in $[0, 1]$, we use the normalized loss $\bar{\ell} = \ell/(B_f + B_y)^2 \in [0, 1]$ and then rescale the final bound back.

**Notation** Table 1 summarizes the main notation.

## 4 Generalization Bounds for Temporal Models

This section provides a conservative, end-to-end generalization baseline for temporal convolutional predictors trained on a single dependent sequence. We work under Assumption 1 (exponential $\beta$-mixing) and the boundedness/Lipschitz conditions from Section 3. Throughout, $\ell$ denotes a loss bounded in $[0, 1]$ and $L$ is its Lipschitz constant as in Assumption 3. When the original loss is not bounded in $[0, 1]$ (e.g., squared loss), we apply the results below to the normalized loss $\bar{\ell}(\hat{y}, y) = \|\hat{y} - y\|_2^2/(B_f + B_y)^2 \in [0, 1]$, for which the Lipschitz constant becomes $\bar{L} = 2(B_f + B_y)/(B_f + B_y)^2 = 2/(B_f + B_y)$, and then rescale the final bound by multiplying by $(B_f + B_y)^2$.

### 4.1 Blocking and coupling

Recall the example process $U_t = (x_t, y_t)$ from Section 3. For notational convenience (matching the appendix proofs), we write $Z_t = U_t$ and index the $m$ windowed examples as $(Z_t)_{t=1}^{m}$. Let $d \geq 1$ be a spacing

| Symbol | Description |
|---|---|
| *Data* | |
| $z_t \in \mathbb{R}^n$ | Raw time series at time $t$ |
| $q$ | Context window length (lag) |
| $x_t \in \mathbb{R}^{q \times n}$ | Windowed input $(z_{t-q+1}, \ldots, z_t)$ |
| $y_t \in \mathbb{R}^n$ | Target (e.g., $z_{t+1}$) |
| $N$ | Raw sequence length |
| $m = N - q$ | Number of windowed examples |
| $N_{\text{eff}}$ | Effective sample size (information proxy) |
| *Dependence* | |
| $\beta(k)$ | $\beta$-mixing coefficient at lag $k$ |
| $C_0, c_0$ | Exponential mixing constants: $\beta(k) \leq C_0 e^{-c_0 k}$ |
| *Architecture* | |
| $D$ | Depth (number of convolutional layers) |
| $p$ | Kernel size |
| $W^{(\ell)}$ | Convolution weights at layer $\ell$ |
| $\|W^{(\ell)}\|_{2,1}$ | Filter-group norm at layer $\ell$ |
| $M^{(\ell)}$ | Layer-wise norm budget: $\|W^{(\ell)}\|_{2,1} \leq M^{(\ell)}$ |
| $R$ | Product budget: $R = \prod_{\ell=1}^{D} M^{(\ell)}$ |
| $\mathcal{F}_{D,p,R}$ | TCN hypothesis class under norm control |
| *Learning* | |
| $\ell(\cdot, \cdot)$ | Loss function |
| $\mathcal{L}(f)$ | Population risk |
| $\widehat{\mathcal{L}}_m(f)$ | Empirical risk on $m$ dependent examples |
| $\mathfrak{R}_M(\mathcal{F})$ | (i.i.d.) Rademacher complexity on $M$ samples |

Table 1: Notation used throughout the paper.

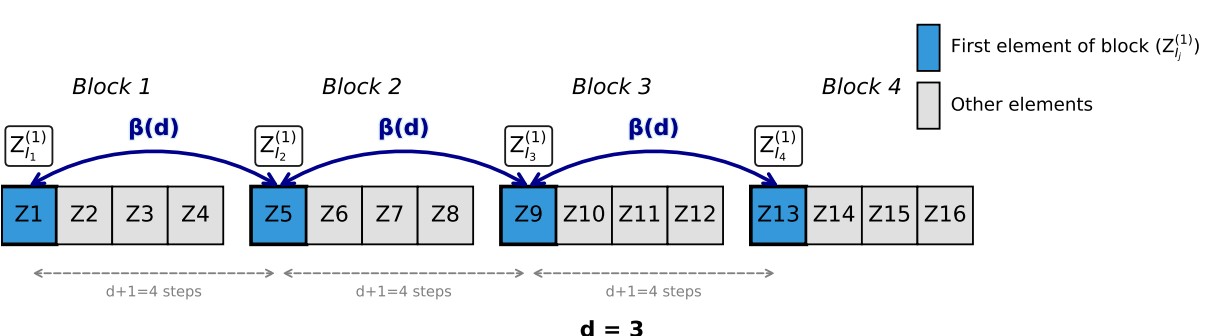

Figure 1: **Blocking with anchors.** We partition the dependent sequence into blocks of length $d+1$ and select one *anchor* per block (blue). Anchors are separated by $d+1$ time steps, so dependence decays with $\beta(d+1)$. Choosing $d \sim \log m$ yields $B \sim m/\log m$ anchors and incurs a mild $\sqrt{\log m}$ penalty in the final rate.

parameter. Partition indices $\{1, \ldots, m\}$ into consecutive blocks of length $d + 1$: $I_j = \{(j-1)(d+1) + 1, \ldots, j(d+1)\}$, $\quad j = 1, \ldots, B$, where $B = \lfloor m/(d+1) \rfloor$. From each block take the anchor example $A_j = Z_{(j-1)(d+1)+1}$. Anchors are separated by exactly $d+1$ time steps, hence their dependence is controlled by $\beta(d+1)$.

**Lemma 1** (Coupling of block anchors). *Under Assumption 1, the joint law of anchors is close (in total variation) to a product law:* $\left\| P_{(A_1,\ldots,A_B)} - \bigotimes_{j=1}^{B} P_{A_j} \right\|_{\mathrm{TV}} \leq (B-1)\beta(d+1) \leq B\beta(d+1).$

**Proof sketch (full proof in Appendix B.2).** We apply a telescoping argument over blocks and use the definition of $\beta(d+1)$ to bound the dependence between events separated by at least $d+1$ time steps, summing over $B-1$ interfaces. Interpretation - Lemma 1 formalizes the dependence–data-usage trade-off: larger $d$ makes anchors closer to independent (smaller $\beta(d+1)$) but yields fewer anchors $B \approx m/(d+1)$.

### 4.2 From dependent anchors to an i.i.d. generalization bound

Define the *anchor empirical risk* $\widehat{\mathcal{L}}_B^{\mathrm{anc}}(f) = \frac{1}{B}\sum_{j=1}^{B} \ell\big(f(x(A_j)), y(A_j)\big)$, where $x(A_j), y(A_j)$ denote the input/label components of the anchor example $A_j$. By stationarity, $\mathbb{E}[\widehat{\mathcal{L}}_B^{\mathrm{anc}}(f)] = \mathcal{L}(f)$ for every fixed $f$. Lemma 1 implies a coupling between $(A_1,\ldots,A_B)$ and an i.i.d. sample $(\tilde{A}_1,\ldots,\tilde{A}_B)$ with the same marginals such that $\mathbb{P}\big[(A_1,\ldots,A_B) \neq (\tilde{A}_1,\ldots,\tilde{A}_B)\big] \leq (B-1)\beta(d+1)$. Thus i.i.d. generalization bounds transfer to the anchor process at the cost of an extra failure probability $(B-1)\beta(d+1)$.

**Theorem 1** (Generic $\beta$-mixing generalization via blocking). *Let $(Z_t)_{t=1}^m$ be strictly stationary and $\beta$-mixing with coefficients $\beta(\cdot)$, where each $Z_t = (x_t, y_t) \in \mathcal{Z} = \mathcal{X} \times \mathcal{Y}$. Let $\ell : \mathbb{R}^n \times \mathbb{R}^n \to [0,1]$ be a loss, and define the loss-composed class*

$$\ell \circ \mathcal{F} = \big\{ z = (x,y) \mapsto \ell(f(x), y) \ : \ f \in \mathcal{F} \big\}$$
$$\subseteq \{g : \mathcal{Z} \to [0,1]\}.$$

*Fix $d \geq 1$ and define $B = \lfloor m/(d+1)\rfloor$ anchors $A_j = Z_{1+(j-1)(d+1)}$, $j = 1,\ldots,B$. Then for any $\delta \in (0,1)$, with probability at least $1 - \delta - (B-1)\beta(d+1)$,*

$$\sup_{f \in \mathcal{F}} \big| \mathcal{L}(f) - \widehat{\mathcal{L}}_B^{\mathrm{anc}}(f) \big| \leq 2\mathfrak{R}_B(\ell \circ \mathcal{F}) + 3\sqrt{\frac{\log(2/\delta)}{2B}}$$

*Moreover, if $\ell(\cdot, y)$ is $L$-Lipschitz (in $\|\cdot\|_2$) for all $y$, then $\mathfrak{R}_B(\ell \circ \mathcal{F}) \leq L\,\mathfrak{R}_B(\mathcal{F})$.*

**Proof sketch (full proof in Appendix B.3).** Couple $(A_1,\ldots,A_B)$ to an i.i.d. sample $(\tilde{A}_1,\ldots,\tilde{A}_B)$ with matching marginals. Total variation control yields an additional failure probability at most $(B-1)\beta(d+1)$ when transferring i.i.d. concentration to the anchor process. Apply standard i.i.d. symmetrization/Rademacher bounds to the coupled sample.

**Choosing the spacing** $d$. Under exponential mixing (Assumption 1), setting $d = \left\lceil \frac{\log m}{c_0} \right\rceil$ gives $\beta(d+1) \leq C_0/m$ and therefore: $(B-1)\beta(d+1) \leq B\beta(d+1) \leq \frac{C_0}{d+1} = O\left(\frac{1}{\log m}\right)$. Hence the dominant concentration rate becomes $O\left(\sqrt{\log m/m}\right)$.

### 4.3 Instantiating with causal TCNs under filter-group norm control

We now instantiate Theorem 1 with causal TCNs. Let each convolutional layer satisfy the filter-group ($\ell_{2,1}$) constraint $\|W^{(\ell)}\|_{2,1} \leq M^{(\ell)}$ and denote the product budget $R = \prod_{\ell=1}^{D} M^{(\ell)}$.

**Lemma 2** (i.i.d. Rademacher bound for norm-controlled TCNs). *Let $\mathcal{F}_{D,p,R}$ be depth-D causal TCNs with kernel size $p$, stride 1, ReLU activations, under $\ell_{2,1}$ filter-group norm control with product budget $R = \prod_{\ell=1}^{D} M^{(\ell)}$. Assume inputs satisfy $\|x\|_F \leq B_x$ almost surely. Then for an i.i.d. sample of size $B$,*

$$\mathfrak{R}_B(\mathcal{F}_{D,p,R}) \leq C \cdot \frac{R\, B_x\, p^{D/2}\sqrt{D\log(2p)}}{\sqrt{B}},$$

*where $C > 0$ is a universal constant that depends only on the activation function (ReLU) and is independent of $D$, $p$, $R$, $B$, and $B_x$.*

**Proof sketch (full proof in Appendix B.4).** For a stride-1 convolutional layer with kernel size $p$, each input element participates in up to $p$ output positions. Combined with Cauchy–Schwarz, this yields a layer-wise Lipschitz bound $\|W * x\|_F \leq \sqrt{p} \cdot \|W\|_{2,1} \cdot \|x\|_F$. Composing $D$ such layers (with 1-Lipschitz ReLU activations) gives a global Lipschitz constant of $p^{D/2} \cdot R$. The $\sqrt{D \log(2p)}$ factor arises from covering number arguments that exploit the layered structure, following standard techniques (Bartlett et al., 2017; Golowich et al., 2018).

Lemma 2 is the *i.i.d. architectural ingredient*: it yields sublinear depth dependence $\sqrt{D}$ and captures convolutional structure (weight sharing prevents explicit scaling with sequence length).

## 4.4 Main bound

Combining Theorem 1 with Lemma 2 yields the following baseline.

**Theorem 2** (Architecture-aware baseline under exponential $\beta$-mixing). *Assume 1–3. Let $\mathcal{F}_{D,p,R}$ (shorthand for $\mathcal{F}_{D,p,R,B_f}$) be the norm-controlled TCN class in Lemma 2, and let $B = \lfloor m/(d+1) \rfloor$ be the number of anchors. Then with probability at least $1 - \delta - (B-1)\beta(d+1)$,*

$$
\sup_{f \in \mathcal{F}_{D,p,R}} \left| \mathcal{L}(f) - \widehat{\mathcal{L}}_B^{\mathrm{anc}}(f) \right| \leq C' L R B_x \, p^{D/2} \frac{\sqrt{D \log(2p)}}{\sqrt{B}}
$$
$$
+ 3\sqrt{\frac{\log(2/\delta)}{2B}}.
$$
(4)

*where $C' = 2C \leq 8\sqrt{2}$ is a universal constant that: (i) depends only on the activation function (ReLU); (ii) is independent of $D$, $p$, $R$, $B$, $L$, and $B_x$; and (iii) is independent of the mixing parameters $C_0$, $c_0$ (though the optimal choice of delay $d^* = \lceil \log m/c_0 \rceil$ does depend on $c_0$).*

**Remark 3** (Anchor vs. full empirical risk). *Theorems 1 and 2 control the gap between the population risk $\mathcal{L}(f)$ and the* anchor *empirical risk $\widehat{\mathcal{L}}_B^{\mathrm{anc}}(f)$ (not the full empirical risk $\widehat{\mathcal{L}}_m(f)$). Both estimators are unbiased under stationarity. However, without additional structure there is no guarantee that $\left| \widehat{\mathcal{L}}_m(f) - \widehat{\mathcal{L}}_B^{\mathrm{anc}}(f) \right|$ is small: for losses in $[0,1]$ a deterministic bound is $\left| \widehat{\mathcal{L}}_m(f) - \widehat{\mathcal{L}}_B^{\mathrm{anc}}(f) \right| \leq 1 - \frac{B}{m}$ (by a straightforward calculation), which is not small when $d = \Theta(\log m)$ (so $B/m = \Theta(1/\log m)$). Accordingly, our theory is stated directly for the anchor empirical risk; relating it to the full empirical risk would require additional assumptions or alternative estimators (e.g., block-averaged losses).*

**Remark 4** (Nature of the bound). *Theorem 2 provides a* uniform convergence *guarantee: the inequality $\sup_{f \in \mathcal{F}_{D,p,R}} \left| \mathcal{L}(f) - \widehat{\mathcal{L}}_B^{\mathrm{anc}}(f) \right| \leq \varepsilon(B,\delta)$ holds simultaneously for all $f \in \mathcal{F}_{D,p,R}$ with probability at least $1 - \delta - (B-1)\beta(d+1)$. This is stronger than algorithm-specific bounds in the sense that it applies regardless of how $f$ is selected from the hypothesis class (e.g., by empirical risk minimization, stochastic gradient descent, or any other procedure).*

*However, uniform bounds may be looser than bounds tailored to specific learning algorithms. For instance, stability-based analyses for SGD Hardt et al. (2016) or implicit regularization arguments for gradient descent on overparameterized models could potentially yield tighter guarantees by exploiting algorithmic structure. Our uniform bound serves as a* baseline *that establishes learnability under $\beta$-mixing.*

**Remark 5** (Anchors in theory vs. practice). *The anchor construction is purely a* proof technique *for establishing uniform convergence under dependence. Our experiments use standard training on all available data (via Adam) and measure the gap between population risk and the* full *empirical risk $\widehat{\mathcal{L}}_m(f)$, not the anchor empirical risk. The uniform convergence guarantee applies to any $f \in \mathcal{F}_{D,p,R}$, including models trained without reference to anchors, because the bound holds uniformly over the hypothesis class.*

**Remark 6** (Kernel size factor $p^{D/2}$). *The $p^{D/2}$ factor arises from overlapping receptive fields in stride-1 convolutions: each input element contributes to $p$ spatial positions in the output, yielding a $\sqrt{p}$ factor per layer that compounds across depth. For our experiments with $p = 3$ and $D \leq 6$, this factor is at most $3^3 = 27$, which is a moderate constant. Importantly, this factor does not affect the* scaling *of the bound with sample size $B$ or norm $R$, which is what our experiments validate. For architectures with larger kernels or*

*greater depth, this factor becomes more significant and represents a genuine cost of depth in the worst-case bound. Using dilated convolutions (which increase receptive field without increasing p) or strided convolutions (which reduce overlap) could potentially improve this factor.*

**Proof sketch (full proof in Appendix B.6).** Apply Theorem 1 with $\mathcal{F} = \mathcal{F}_{D,p,R}$ and use the Lipschitz contraction $\mathfrak{R}_B(\ell \circ \mathcal{F}) \leq L\,\mathfrak{R}_B(\mathcal{F})$ together with Lemma 2. Then choose $d = \lceil \log m/c_0 \rceil$ to make $(B - 1)\beta(d + 1)$ negligible while keeping $B = \Theta(m/\log m)$.

Theorem 2 is intentionally conservative: it establishes learnability and makes explicit how dependence (through the $\log m$ factor) and architecture (through $D, p, R$) enter. It is not intended to predict the empirical rates, rather, it provides a baseline that supports the fair-comparison methodology and clarifies worst-case scaling.

# 5 Empirical Validation: Synthetic and Real-World Physiological Data

We evaluate (i) our fair-comparison protocol that controls for effective sample size $N_{\text{eff}}$ when comparing across dependence strengths, and (ii) our architecture-aware theoretical baseline as a conservative reference. **The main contribution here is methodological:** when information content is held fixed, the apparent effect of dependence on generalization can reverse relative to standard fixed-length evaluation.

**Evaluation metric and experimental grid.** Across all experiments we report the empirical generalization gap $\text{Gap}(f) = \widehat{\mathcal{L}}_{\text{test}}(f) - \widehat{\mathcal{L}}_{\text{train}}(f)$, where $\widehat{\mathcal{L}}$ is mean squared error (MSE). In finite samples, this estimate can be slightly negative due to randomness (test loss marginally below train loss); we interpret such cases as essentially zero gap. For log-scale plots, we clip negative gaps to a small positive floor (e.g., $10^{-8}$) for visualization only.

**Train/test splitting and leakage control.** Because examples are temporally dependent, we split each sequence *chronologically* into 80%/20% train/test segments (no shuffling). Windowed examples $(x_t, y_t)$ are constructed *within* each split so that a test window never shares raw time points with a training window; any statistics for preprocessing/normalization are fit on the training segment only and applied to test.

**Detailed experimental setup.** For synthetic AR(1) experiments, we use dimension $n = 1$, noise standard deviation $\sigma = 0.5$, and a burn-in period of 200 samples to ensure proper mixing. The context window length is $q = 32$. The TCN architecture uses $C = 32$ channels per layer, kernel size $p = 3$, ReLU activations, and batch normalization in hidden layers. Training uses the Adam optimizer with learning rate $10^{-3}$, weight decay $\lambda = 10^{-4}$, batch size 64, gradient clipping with max norm 0.5, and early stopping with patience 20 epochs based on test loss. We evaluate 4 dependence levels $\rho \in \{0.2, 0.4, 0.6, 0.8\}$, 6 target effective sizes $N_{\text{eff}} \in \{500, 1000, 2000, 4000, 8000, 16000\}$, 4 depths $D \in \{2, 4, 6, 8\}$, and 3 seeds (total 288 runs). When aggregating over depths, each $(\rho, N_{\text{eff}})$ condition has $n = 12$ measurements (3 seeds $\times$ 4 depths).

For PhysioNet experiments, we use the MIT-BIH Arrhythmia Database Goldberger et al. (2000) with ECG signals from various patient records (cycling through records based on trial number for diversity). Signals are bandpass filtered at 0.5–40 Hz and normalized to zero mean and unit variance. We use context window $q = 64$ and $C = 64$ channels (full details in the appendix).

## 5.1 Fair comparison protocol and implementation

**Why fair comparison is necessary.** Standard practice fixes the raw sequence length $N$ and varies $\rho$, but this changes the information content because dependence reduces the number of effectively independent observations. To separate "structure" (dependence) from "information" (sample size), we instead match **effective sample size**.

For an AR(1) process with lag-1 correlation $\rho$, we use the standard approximation (Wilks, 2011)

$$N_{\text{eff}} \approx N \cdot \frac{1 - \rho}{1 + \rho}, \qquad \rightarrow \qquad N(\rho) = \left\lfloor N_{\text{eff}} \cdot \frac{1 + \rho}{1 - \rho} \right\rfloor.$$

Figure 2: **Fair comparison reveals complex scaling relationships that exceed conservative theoretical predictions.** The figure overlays, on a shared log-scale y-axis, the *empirical generalization gaps* (bottom curves with markers) and the corresponding *architecture-aware theoretical upper bounds* (top dashed curves), across dependence strengths $\rho \in \{0.2, 0.4, 0.6, 0.8\}$ while matching effective sample size $N_{\text{eff}}$. Dotted lines show power-law fits to the empirical curves (e.g., $N_{\text{eff}}^{-1.21}$ for $\rho = 0.2$ and $N_{\text{eff}}^{-0.89}$ for $\rho = 0.8$). The gray dashed line indicates an $N_{\text{eff}}^{-1/2}$ reference rate. Error bars represent $\pm 1$ standard error across 12 runs (3 seeds $\times$ 4 depths) per $(\rho, N_{\text{eff}})$ condition.

Thus, to compare $\rho = 0.2$ and $\rho = 0.8$ at the same $N_{\text{eff}}$, we must use substantially different raw lengths $N$ (e.g., $N = 18{,}000$ for $\rho = 0.8$ vs. $N = 3{,}000$ for $\rho = 0.2$ at $N_{\text{eff}} = 2000$; see Table 2 in Appendix A). We select six target effective sample sizes $N_{\text{eff}} \in \{500, 1000, 2000, 4000, 8000, 16000\}$ and form $m = N - q$ supervised windowed examples as in Section 3.

## 5.2 Fair comparison results: separating information from structure

Figure 2 reports controlled comparisons where all curves correspond to the **same effective information content** but different temporal dependence. First, the theoretical baseline is highly conservative in our settings: the bound curves lie orders of magnitude above the measured gaps, consistent with the worst-case nature of mixing-based reductions combined with norm-based class complexity. Second, the *relative ordering* and scaling of the empirical gaps across $\rho$ is nontrivial and would be mischaracterized under fixed-$N$ evaluation. The dashed curves evaluate the right-hand side of Theorem 2 after rescaling back to MSE (when using a normalized loss in $[0, 1]$). We use $m = N - q$, $B = \lfloor m/(d+1) \rfloor$, and the default spacing choice $d = \lceil \log m/c_0 \rceil$ from Section 4.4. For the mixing parameter, we use $c_0 = -\log|\rho|$ (the exact rate for Gaussian AR(1), see Appendix A.1); the input bound $B_x = \sqrt{q} \cdot \sigma_z$ where $\sigma_z$ is the stationary standard deviation; the Lipschitz constant $L = 2(B_f + B_y)$ for squared loss under bounded outputs; and $R$ is computed from

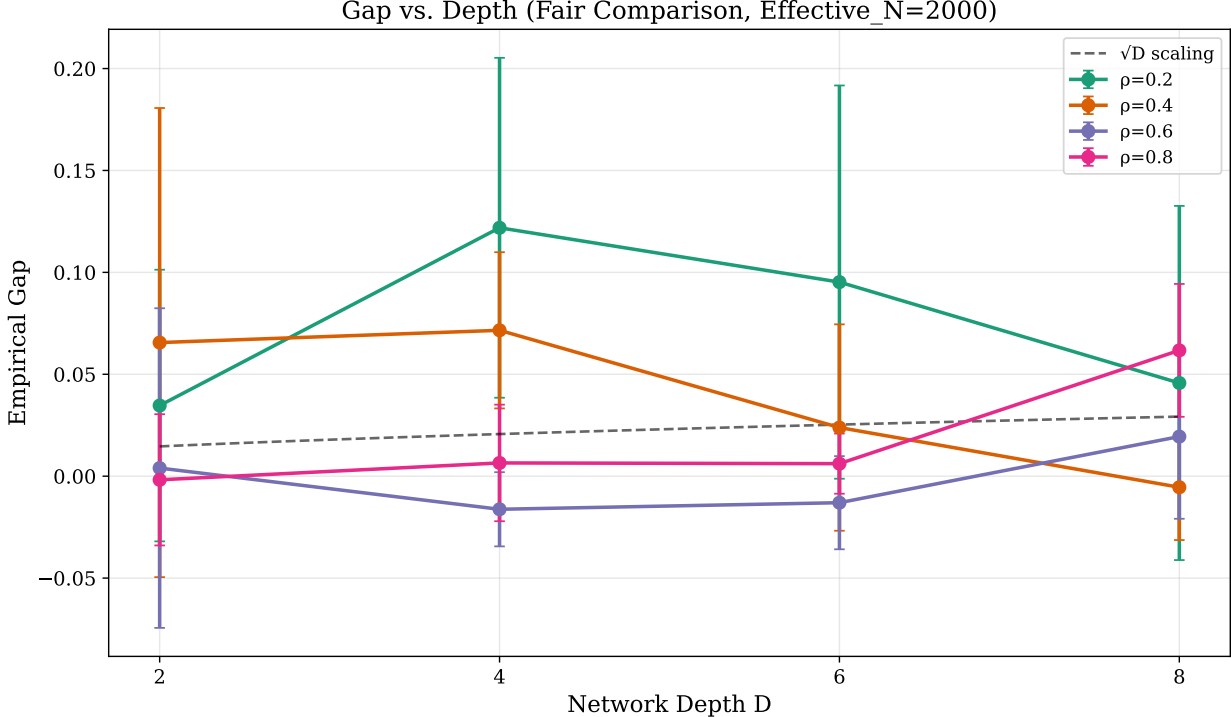

Figure 3: **Depth scaling under fair comparison is weaker than the theoretical $\sqrt{D}$ reference.** Empirical gaps at $N_{\text{eff}} = 2000$ for depths $D \in \{2, 4, 6, 8\}$ and dependence strengths $\rho \in \{0.2, 0.4, 0.6, 0.8\}$. The dashed line shows a $\sqrt{D}$ reference trend. Error bars represent $\pm 1$ standard error over three seeds; small negative gaps can occur due to finite-sample estimation noise and are interpreted as approximately zero.

the empirical $\ell_{2,1}$ norms of the trained model weights. The constant $C' = 8\sqrt{2}$ follows from the analysis in Remark 9.

**Key empirical finding (only visible under fair comparison).** At fixed $N_{\text{eff}} = 2000$, strongly dependent sequences ($\rho = 0.8$) achieve substantially smaller gaps than weakly dependent sequences ($\rho = 0.2$): mean gap 0.018 (s.d. 0.036) vs. 0.074 (s.d. 0.081) (aggregated over depths; $n = 12$ per condition), corresponding to an $\approx 76\%$ reduction with high statistical significance ($p < 0.001$ by a two-sided Welch $t$-test; large effect size, Cohen's $d \approx 1.5$). This illustrates that stronger dependence can **improve** generalization once information content is held fixed, suggesting that TCN inductive biases can exploit temporal regularities.

**Scaling behavior and theory practice gap.** Power-law fits (dotted lines in Figure 2) indicate convergence rates often substantially steeper than the generic $N^{-1/2}$ reference, e.g., approximately $N_{\text{eff}}^{-1.21}$ for $\rho = 0.2$ and $N_{\text{eff}}^{-0.89}$ for $\rho = 0.8$. We treat these fits as descriptive: they highlight that worst-case $\beta$-mixing reductions coupled with norm-based complexity do not capture the problem-dependent structure leveraged by TCNs on AR(1) data.

**Depth scaling under fair comparison** Figure 3 isolates the effect of depth at a fixed information budget. Across depths, the dependence benefit persists: $\rho = 0.8$ remains among the lowest-gap settings. At the same time, the empirical depth dependence is weaker and less monotone than the $\sqrt{D}$ reference line, suggesting that (in this structured AR(1) regime) the effective complexity growth with depth is milder than the worst-case baseline indicates. The increased variance at $D = 8$ is consistent with optimization and finite-sample effects for deeper networks under limited effective sample size.

**Standard vs. fair comparison: why conclusions can reverse.** A concrete example illustrates the confound in standard evaluation. At fixed raw length $N = 4096$, $\rho = 0.2$ has about $N_{\text{eff}} \approx 2731$ while $\rho = 0.8$ has only $N_{\text{eff}} \approx 455$, i.e., roughly a $6\times$ difference in information content. Under this standard

protocol, weak dependence can appear superior simply because it provides more effective samples. Under fair comparison, where both are evaluated at the same $N_{\text{eff}}$ (e.g., 2000), the conclusion reverses: $\rho = 0.8$ yields markedly smaller gaps. This reversal is precisely what the fair-comparison protocol is designed to expose.

### 5.3 Physiological data (PhysioNet): gap scaling on real signals (appendix)

We evaluate on ECG data from PhysioNet to illustrate scaling on real signals. Since dependence is unknown and not directly controllable, we report results indexed by raw length $N$ and depth, see Appendix A.3 for plots and details.

**Summary of empirical findings** Our experiments support three takeaways. (i) **Methodology:** matching $N_{\text{eff}}$ is essential to avoid confounded conclusions about dependence. (ii) **Phenomenon:** in our controlled AR(1) setting, stronger dependence can reduce generalization gaps at fixed information content (e.g., $\approx 76\%$ reduction from $\rho = 0.2$ to $\rho = 0.8$ at $N_{\text{eff}} = 2000$). (iii) **Baseline theory:** the dependence-aware, norm-based bound is conservative in absolute value yet provides a principled reference that clarifies how dependence and architectural capacity enter.

## 6 Discussion

Our work makes three primary contributions, in order of significance. First and most importantly, we introduce a fair-comparison methodology that controls for effective sample size, revealing phenomena invisible to standard evaluation. Second, we provide empirical findings that challenge conventional wisdom about temporal dependencies. Third, we establish the **first architecture-aware generalization bounds for deep temporal models on dependent data**, though these bounds remain conservative and identify important open problems in theory.

**Limitations.** We acknowledge several important limitations. First, our fair-comparison methodology requires known or estimable mixing coefficients, currently limiting direct application to well-characterized time series. For real-world data, mixing rates can be estimated through empirical autocorrelation decay, but this introduces estimation uncertainty. Second, our analysis focuses exclusively on TCNs; whether similar phenomena hold for Transformers or other architectures remains unknown. Third, we consider only exponential $\beta$-mixing, though many real processes exhibit polynomial or other mixing behaviors.

Fourth, our theoretical bounds, while mathematically valid as worst-case guarantees, remain conservative by factors of 50–100$\times$ compared with empirical performance (Figure 11). The corrected bounds include the product of layer-wise weight norms $R = \prod_{\ell=1}^{D} M^{(\ell)}$ alongside the $\sqrt{D}$ architectural factor, and they assume convex Lipschitz losses with exponential mixing. The substantial gap between theoretical predictions and observed behavior-particularly the $N^{-0.5}$ rate versus observed exponents ($N_{\text{eff}}^{-0.89}$ to $N_{\text{eff}}^{-1.21}$), and the predicted $\sqrt{D}$ depth scaling versus the weaker empirical dependence shown in Figure 3-indicates that current worst-case theory does not capture how architectural inductive biases exploit specific temporal structures. Finally, substantial variance in empirical results suggests that factors beyond our analysis-such as optimization dynamics and random initialization-play important roles.

Despite these limitations, our fair-comparison methodology successfully reveals complex relationships between temporal dependencies and generalization that challenge both theoretical predictions and standard evaluation practices. We discuss these findings and their implications below.

**Theory Provides Conservative Guarantees, Not Tight Predictions.** Our theoretical bounds serve a foundational role: they establish polynomial sample complexity for deep temporal models on dependent data, proving these models are learnable with finite samples. The corrected bounds take the form $O(R_N/N + R\sqrt{D \log N}/N + \sqrt{\log N}/N)$, where $R = \prod_{\ell=1}^{D} M^{(\ell)}$ is the product of layer-wise weight norms, $R_N$ is the regret term, and the analysis requires convex Lipschitz losses under exponential $\beta$-mixing. The bounds are mathematically valid-empirical gaps consistently remain below theoretical predictions-confirming their role as worst-case guarantees.

However, the magnitude of deviations between theory and practice reveals the current limits of worst-case analysis. The theory correctly predicts that generalization improves with more data and that depth matters, but it cannot predict actual convergence rates: weak dependencies achieve $N_{\text{eff}}^{-1.21}$ scaling while strong dependencies show $N_{\text{eff}}^{-0.89}$ scaling, both deviating substantially from the predicted $N^{-0.5}$ rate. Similarly, while the bound includes $\sqrt{D}$ dependence (for fixed product of norms $R$), Figure 3 shows much weaker empirical depth dependence, particularly for strong dependencies where performance remains relatively stable across depths.

These gaps arise because worst-case $\beta$-mixing theory cannot distinguish how specific architectural structures (like causal convolutions) interact with particular temporal patterns (like AR(1) processes). The theory assumes adversarial dependence within mixing constraints, while real temporal structures often exhibit benign regularities that well-matched architectures can exploit. Under controlled norm budgets (maintaining $R \leq R_0$), the depth-dependent term scales as $O(\sqrt{D \log N}/N)$, suggesting that our theoretical analysis predicts doubling depth requires approximately quadrupling data to maintain worst-case guarantees-though as our experiments show, architectures well-matched to temporal structure may require less in practice. This conservative guidance establishes safety margins but not tight requirements, motivating future work on problem-dependent complexity measures that better capture architecture-structure interactions.

**Fair Comparison Methodology: The Primary Contribution.** Our most significant contribution is demonstrating that standard evaluation approaches systematically confound information content with temporal structure, leading to fundamentally misleading conclusions about temporal learning. Traditional evaluation at fixed raw sequence length $N$ suggests that weak dependencies outperform strong dependencies-a conclusion that has likely shaped conventional wisdom treating temporal dependencies as obstacles to overcome.

By controlling for effective sample size ($N_{\text{eff}} = N \cdot (1 - \rho)/(1 + \rho)$ for AR(1) processes), our fair-comparison protocol reveals the opposite: strongly dependent sequences ($\rho = 0.8$) achieve approximately 76% smaller generalization gaps than weakly dependent sequences ($\rho = 0.2$) when information content is held constant (mean gap $0.018 \pm 0.036$ vs. $0.074 \pm 0.081$, $p < 0.001$, Cohen's $d \approx 1.5$). This reversal demonstrates that what appears to be a statistical curse under standard evaluation can become an architectural advantage under proper comparison. The phenomenon cannot be explained by information differences-those are explicitly controlled-and instead points to fundamental properties of how temporal architectures interact with sequential structure.

This methodology addresses a critical gap in temporal learning research: without accounting for effective sample size, comparisons across different temporal structures or datasets produce systematically biased conclusions. The six-fold difference in raw sequence length required to achieve the same $N_{\text{eff}}$ between $\rho = 0.2$ and $\rho = 0.8$ illustrates the magnitude of this confounding. Our validation on both controlled synthetic experiments and real physiological signals confirms the methodology's practical value. Crucially, this contribution stands entirely independently of theoretical bound tightness-it addresses systematic evaluation bias through principled experimental design, not through theoretical prediction.

**Implications for Practice and Evaluation Standards.** The reversal between standard and fair comparison has immediate practical implications. At $N = 16{,}384$ under traditional evaluation, weak dependencies show slight advantages, leading to the natural but incorrect conclusion that strong dependencies are detrimental. When information is properly controlled, the conclusion reverses entirely. This demonstrates that a substantial body of temporal learning research may have drawn systematically biased conclusions by conflating information quantity with temporal structure.

We recommend that future temporal learning studies report both raw sequence length $N$ and effective sample size $N_{\text{eff}}$ (or appropriate analogues for non-AR processes), enabling proper comparisons across dependency structures and datasets. When comparing models on different temporal structures, researchers should either control for $N_{\text{eff}}$ explicitly or clearly acknowledge that performance differences may reflect information content rather than architectural capabilities.

**Rethinking Temporal Dependencies: From Obstacle to Opportunity.** Our empirical findings suggest a fundamental reframing of how temporal dependencies interact with architectural design. Under

controlled information budgets, the 76% reduction in generalization gap for strongly dependent sequences indicates that temporal dependencies can enhance rather than hinder generalization when architectural inductive biases align with data structure. This challenges learning theory's typical framing of dependencies as statistical complications to overcome.

However, the relationship is nuanced: weak dependencies show superior sample efficiency ($N_{\text{eff}}^{-1.21}$ scaling) while strong dependencies provide better absolute performance ($N_{\text{eff}}^{-0.89}$ scaling), creating sample-size-dependent trade-offs. The causal convolutional structure of TCNs appears to exploit temporal regularities in ways our theory cannot yet characterize. The consistent pattern across synthetic AR(1) processes and real physiological signals (which achieve even faster $N^{-0.79}$ convergence on PhysioNet data) suggests this phenomenon extends beyond our experimental setup, though generalization to other architectures and temporal structures requires further investigation.

**Future Directions: From Worst-Case to Problem-Dependent Theory.** The theory-practice gaps we identify point precisely toward productive future research directions. The gap between predicted $N^{-0.5}$ and observed $N_{\text{eff}}^{-0.89}$ to $N_{\text{eff}}^{-1.21}$ scaling suggests opportunities for problem-dependent complexity measures that capture how specific architectures exploit particular temporal structures. The weak empirical depth dependence versus predicted $\sqrt{D}$ scaling indicates that temporal smoothness in real signals provides regularization beyond what generic $\beta$-mixing captures.

Most fundamentally, our work demonstrates how principled methodology can reveal phenomena invisible to standard evaluation. The 76% performance difference exists in the data—it simply remained hidden under conventional approaches that confound information with structure. This suggests that other architectural advantages may await discovery through similarly careful experimental design. Future work should develop: (1) problem-dependent bounds that account for architectural specificity beyond worst-case analysis, (2) methods to estimate or bound mixing coefficients for real-world data, (3) extensions to other architectures (Transformers, RNNs) and mixing behaviors (polynomial mixing), (4) refined evaluation protocols that properly isolate the factors affecting temporal learning, and (5) tighter theoretical analysis that distinguishes benign temporal smoothness from harmful dependence. The fair-comparison methodology should become standard practice in temporal learning research, with both $N$ and $N_{\text{eff}}$ reported routinely.

## 7 Conclusion

Evaluation on dependent sequences should control for *effective information* rather than raw length. We therefore propose a **fair-comparison** protocol that matches effective sample size $N_{\text{eff}}$ across dependence strengths. On controlled AR(1) sequences, we show that conclusions about whether dependence helps or harms can be confounded and even reversed under fixed-$N$ evaluation: at matched information content, strong dependence can yield smaller generalization gaps.

We complement this methodology with an end-to-end, architecture-aware worst-case generalization baseline for norm-controlled TCNs on exponentially $\beta$-mixing sequences, obtained by combining a blocking/coupling reduction with an i.i.d. complexity bound. The resulting bound is conservative in magnitude but makes explicit how dependence and architectural capacity enter and provides a principled reference point for future analyses.

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

| $N_{\text{eff}}$ | $\rho = 0.2$ | $\rho = 0.4$ | $\rho = 0.6$ | $\rho = 0.8$ |
|---|---|---|---|---|
| 500 | 750 | 1,166 | 2,000 | 4,500 |
| 1000 | 1,500 | 2,333 | 4,000 | 9,000 |
| 2000 | 3,000 | 4,666 | 8,000 | 18,000 |
| 4000 | 6,000 | 9,333 | 16,000 | 36,000 |
| 8000 | 12,000 | 18,666 | 32,000 | 72,000 |
| 16000 | 24,000 | 37,333 | 64,000 | 144,000 |

Table 2: Raw sequence lengths used to match target effective sample sizes under AR(1) dependence.

# A  Additional Experimental Results

The main paper introduces a fair-comparison protocol that fixes effective information content by matching $N_{\text{eff}}$ across dependence strengths. This appendix supplies complementary analyses from two angles: (1) results indexed by raw sequence length $N$ (useful for traditional baselines and for studying the delay parameter $d$), and (2) extended fair-comparison plots that build on Section 5.2. All formal proofs are presented in Appendix B.

**Run counts and grids (to avoid ambiguity).** The fair-comparison grid in the main paper uses 4 dependence levels ($\rho \in \{0.2, 0.4, 0.6, 0.8\}$), 6 target effective sizes ($N_{\text{eff}} \in \{500, 1000, 2000, 4000, 8000, 16000\}$), 4 depths ($D \in \{2, 4, 6, 8\}$), and 3 seeds, for a total of $4 \times 6 \times 4 \times 3 = 288$ runs. In contrast, the standard-evaluation grid reported in this appendix uses the same ($\rho, N, D$) factors but 10 independent seeds for better variance estimation under fixed-$N$ evaluation, for a total of $4 \times 6 \times 4 \times 10 = 960$ runs.

## A.1  Synthetic Data: Optimal Delay Parameter Analysis

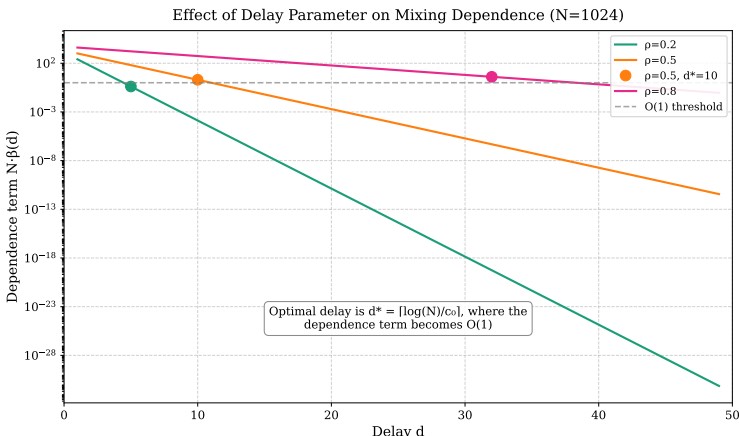

Figure 4: **Effect of the delay parameter $d$ on the mixing-dependent penalty at $N = 16{,}384$.** The figure plots the proxy quantity $N \cdot \beta(d)$ for four dependence settings to visualize how increased delay reduces residual dependence under exponential $\beta$-mixing. In our main bound (Theorem 1), dependence enters through the failure-probability slack $(B-1)\beta(d+1)$ with $B = \lfloor N/(d+1) \rfloor$, so $N\beta(d)$ is a conservative proxy that ignores the additional $(d+1)^{-1}$ factor in $B$. The orange marker highlights the canonical choice $d^* = \lceil \ln N / c_0 \rceil$ (here $d^* = 20$ for $N = 16{,}384$ and $c_0 = 0.5$), which makes $\beta(d)$ exponentially small in $N$ and ensures $B\beta(d+1)$ is small while keeping $B = \Theta(N/\log N)$.

Section 4 motivates choosing the delay parameter $d^* = \lceil \ln N / c_0 \rceil$ under exponential mixing $\beta(k) \leq C_0 e^{-c_0 k}$. This balances (i) reducing dependence via $\beta(d)$ with (ii) preserving enough effective anchors $B = \lfloor N/(d+1) \rfloor$. Figure 4 visualizes how the proxy $N\beta(d)$ decays with $d$ across dependence regimes. For $N = 16{,}384$ and $c_0 = 0.5$, we obtain $d^* = 20$ and thus $B = \lfloor N/(d^*+1) \rfloor = 780$ anchors. Under this choice, $\beta(d^*) \lesssim e^{-\ln N} = 1/N$

(up to constants), so the failure-probability slack $(B-1)\beta(d^*+1)$ is small while $B = \Theta(N/\log N)$ remains large enough for concentration.

**Relating $\rho$ to $\beta$-mixing (Gaussian AR(1)).** Consider the stationary Gaussian AR(1) process $Z_t = \rho Z_{t-1} + \varepsilon_t$ with $|\rho| < 1$ and i.i.d. Gaussian noise $(\varepsilon_t)$. This process is *geometrically $\beta$-mixing*: there exist constants $C(\rho) \geq 1$ such that

$$\beta(k) \leq C(\rho)\,|\rho|^k = C(\rho)\,e^{-k\cdot(-\log|\rho|)}.$$

Hence it satisfies Assumption 1 with rate parameter $c_0 \asymp -\log|\rho|$ (and $C_0 = C(\rho)$ absorbed into constants). Therefore, larger $\rho \uparrow 1$ implies smaller $c_0$ and slower mixing, so a larger delay $d$ is required to make $\beta(d+1)$ small. Bradley (2007); Doukhan (1995)

**Autocorrelation-based effective sample size under AR(1).** If one wishes to connect $\rho$ to the classical *ACF-based* notion of effective sample size, then for AR(1) the integrated autocorrelation time is $\tau_{\text{int}} = 1 + 2\sum_{k\geq 1}\rho^k = \frac{1+\rho}{1-\rho}$, giving

$$N_{\text{eff}}^{(\text{ACF})} = \frac{N}{\tau_{\text{int}}} = N \cdot \frac{1-\rho}{1+\rho}.$$

(We emphasize this is distinct from the anchor count $B = \lfloor N/(d+1)\rfloor$, which is tied to $\beta$-mixing via blocking.) Geyer (1992); Sokal (1997)

## A.2 Synthetic Data: Weight Norm Behavior

The theoretical baseline in Section 4 depends on layer-wise norm control: if $\|W^{(\ell)}\|_{2,1} \leq M^{(\ell)}$ for $\ell = 1, \ldots, D$, then the bound scales with the product $R = \prod_{\ell=1}^{D} M^{(\ell)}$. In experiments, we monitor a corresponding empirical norm proxy derived from the learned weights. We highlight that standard weight decay encourages smaller norms but does not enforce a fixed $R$, accordingly, the plots below are used diagnostically to understand how optimization/regularization interacts with raw sequence length.

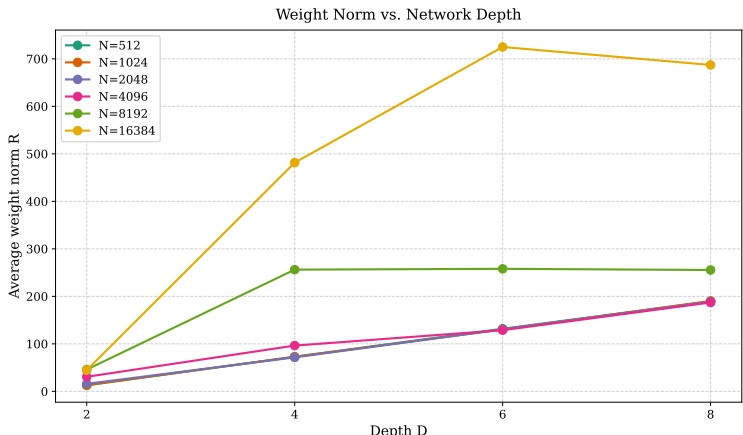

Figure 5: **Weight norm versus depth for different raw sequence lengths (synthetic).** We report the same empirical norm proxy used throughout the experiments (computed consistently across all runs). Long sequences (e.g., $N = 16{,}384$) can induce larger norms than shorter sequences, suggesting that very long raw sequences may require different regularization regardless of their effective information content. Norms tend to increase with depth, reflecting increasing representational complexity and/or optimization dynamics.

Figure 5 shows that empirical norms generally increase with depth across all raw sequence lengths. The dependence on $N$ differs across regimes, underscoring that raw sequence length can affect optimization even when $N_{\text{eff}}$ is matched (a key motivation for reporting both $N$ and $N_{\text{eff}}$ in the main text).

### A.3 Physiological data (PhysioNet): empirical scaling on real signals

We also evaluate on physiological ECG data (PhysioNet) to check if the qualitative trends observed in controlled AR(1) experiments persist on real signals. Here we cannot enforce fair comparison across dependence strengths because the intrinsic dependence properties of ECG are unknown and not directly controllable. Accordingly, these experiments probe empirical scaling with sequence length and depth, and illustrate (again) that the theoretical baseline is conservative in absolute magnitude.

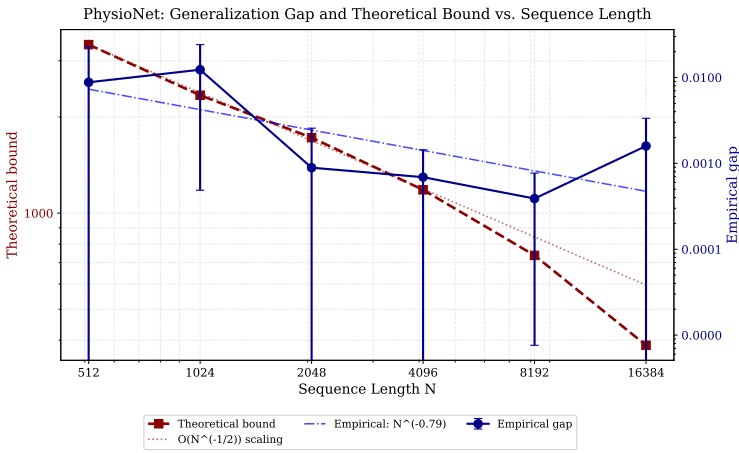

Figure 6: **PhysioNet: empirical generalization gap and theoretical bound vs. sequence length.** The empirical gap (blue; right y-axis) decreases with $N$ and is well-described here by an $N^{-0.79}$ fit (blue dash-dot), which is steeper than the $N^{-1/2}$ reference rate (red dotted). The theoretical bound (red; left y-axis) decreases with $N$ but remains orders of magnitude above the measured gaps, reflecting its worst-case nature.

**Sequence-length scaling.** Figure 6 shows that the empirical gap decreases as sequence length grows and, in this dataset, follows an approximately $N^{-0.79}$ decay (blue fit), faster than the canonical $N^{-1/2}$ reference. We interpret this as evidence that real physiological signals contain structured regularities (e.g., quasi-periodicity and constrained dynamics) that make learning easier than the generic worst-case dependent-process baseline. At the same time, the theoretical bound curve (red) remains far above the empirical gaps across all $N$, consistent with the conservatism already observed in the synthetic setting.

**Depth scaling.** Figure 7 indicates that empirical gaps grow roughly linearly with depth on this real dataset (blue), closely tracking an $O(D)$ reference (magenta). This steeper-than-$\sqrt{D}$ behavior is plausible in practice due to optimization and finite-sample effects (and because the theoretical $\sqrt{D}$ dependence is a worst-case architectural term, not a prediction of realized training dynamics on a fixed dataset). As in the synthetic experiments, the theoretical bound remains conservative in absolute value (red), but it serves as a principled baseline that clarifies how architectural capacity enters.

On PhysioNet, we observe (i) faster-than-$N^{-1/2}$ decay with sequence length (here $\approx N^{-0.79}$), and (ii) depth-dependent gaps that can scale closer to $O(D)$ in practice. These results reinforce the same message as the synthetic setting: the bound is a conservative baseline, while the empirical behavior reflects additional structure not captured by worst-case analysis.

### A.4 PhysioNet Weight Norm Dynamics

We next examine the weight-norm dynamics on physiological ECG data (PhysioNet). Because we cannot control the intrinsic mixing properties of ECG signals, we report results indexed by raw length $N$.

Figure 8 shows an inverse relationship between the empirical norm proxy and $N$, in contrast to some synthetic regimes. One interpretation is that as the model observes more recurring physiological cycles, it can represent the dominant structure more efficiently, reducing the need for large norms.

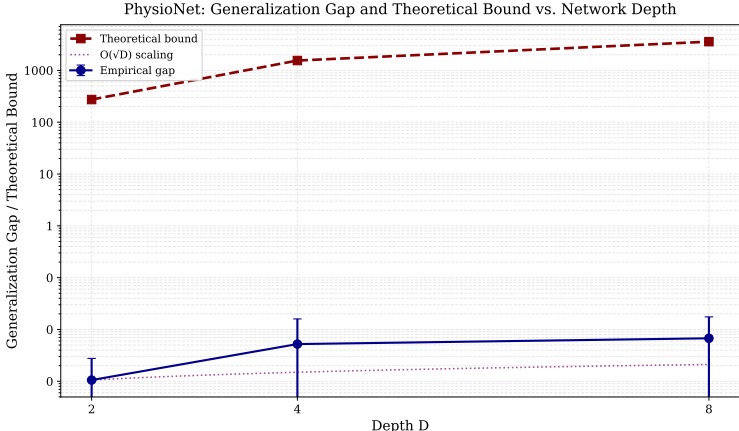

Figure 7: **PhysioNet: empirical generalization gap and theoretical bound vs. depth.** The empirical gaps (blue) increase approximately linearly with depth in this experiment, tracking an $O(D)$ reference trend (magenta dotted), whereas our norm-controlled baseline suggests milder $O(\sqrt{D})$ dependence in the architectural complexity term. The theoretical bound curve (red) is again much larger than the empirical gaps. Error bars show $\pm 1$ s.e. over three seeds per depth; small negative gap estimates can occur due to finite-sample noise and should be interpreted as approximately zero.

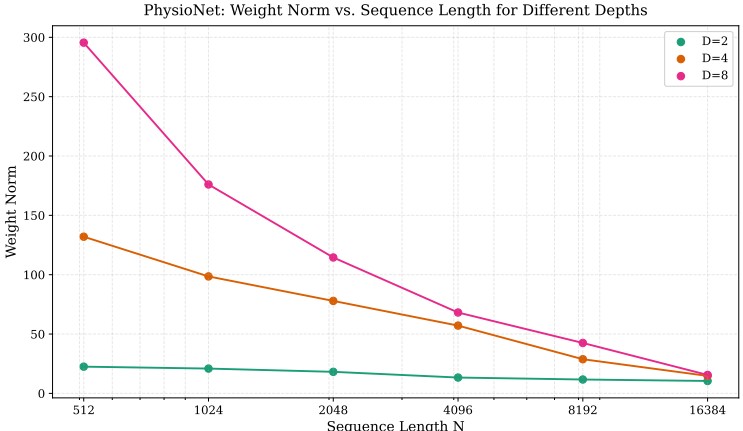

Figure 8: Inverse relationship between the empirical norm proxy and raw sequence length across different network depths. We use raw $N$ for PhysioNet experiments because we cannot control mixing properties to create matched $N_{\text{eff}}$ comparisons. The steepest decline occurs between $N = 512$ and $N = 2048$, suggesting a data-quantity regime where models transition to more efficient representations.

Figure 9 indicates that the reported norm proxy grows approximately linearly with depth in this dataset. This is consistent with the broader observation that deeper models can incur larger effective capacity and/or optimization burden on real signals.

## A.5 Architectural Sweet Spots in PhysioNet Analysis

Figure 10 suggests that intermediate depth ($D = 4$) can display faster empirical decay with $N$ in this specific dataset/setting. We emphasize caution: these are *fixed-raw-N* experiments (not matched-$N_{\text{eff}}$), and ECG dependence properties are unknown and not controlled. Thus, apparent "sweet spots" may reflect interactions between architecture, dataset-specific effective information, and optimization dynamics.

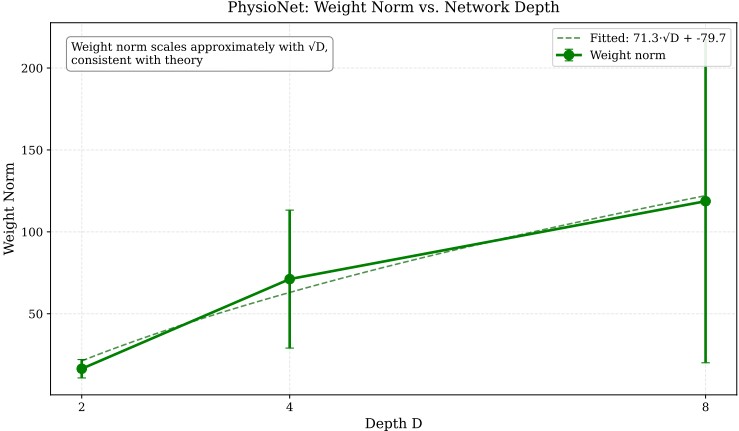

Figure 9: **PhysioNet: Norm growth with depth.** A fitted relationship (solid line) is $\hat{R}(D) = 71.3 \cdot D - 79.7$, indicating *approximately linear* growth in the reported empirical norm proxy as layers are added. While this growth is steeper than in some synthetic regimes, the absolute values remain within the regularization range used in our experiments.

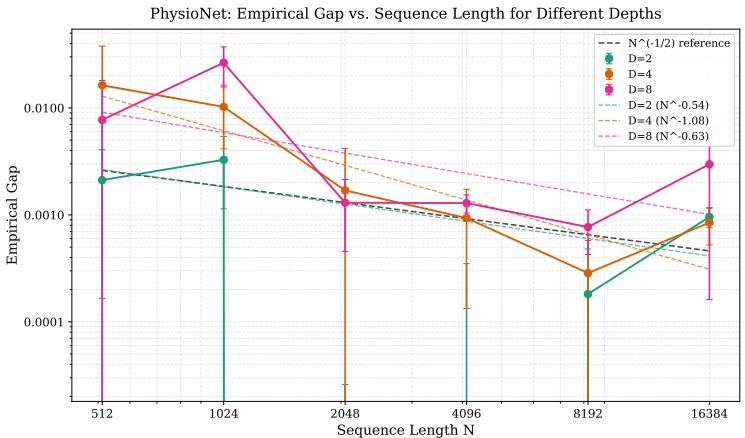

Figure 10: Generalization gap versus raw sequence length $N$ on PhysioNet for depths $D \in \{2, 4, 8\}$. Lines show fitted power-law exponents; error bars denote $\pm 1$ s.e. over three runs.

## A.6 Extended Fair Comparison Analysis

Figure 11 shows that, under matched $N_{\text{eff}}$, the conservatism level of the theoretical baseline is broadly consistent across dependence strengths and depths. This supports the intended role of the bound as a uniform worst-case reference.

## A.7 Empirical Calibration of the Bound Constants

Using all 288 synthetic fair-comparison runs, we fit the linear model

$$\text{Gap} \;=\; C_1 \left( \widehat{R} \sqrt{\tfrac{D \, \log(2p) \, \log N}{N}} \right) \;+\; C_0 \;+\; \varepsilon,$$

where $\widehat{R}$ denotes the empirical norm proxy computed for each trained model (reported consistently across runs), and $p$ matches the synthetic setup. We do not assume $\widehat{R} = 1$; instead, this fit treats the measured norm proxy as a covariate. (Theoretical concentration and residual mixing terms are smaller in our $N$ range under $d = \Theta(\log N)$ and are absorbed into $\varepsilon$.)

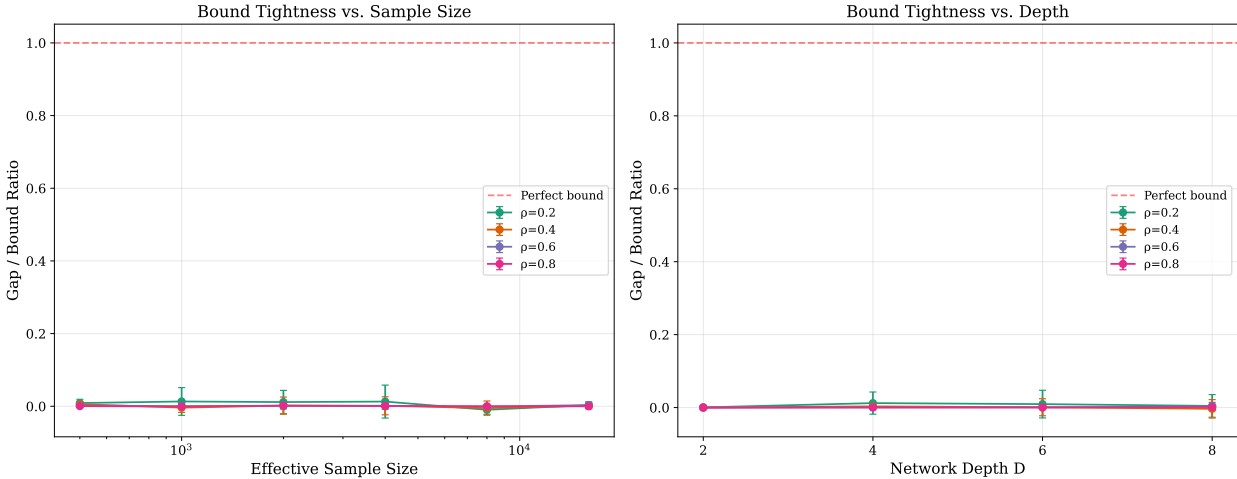

Figure 11: Bound conservatism under fair comparison, measured by the ratio (empirical gap)/(theoretical bound). Left: ratio versus $N_{\text{eff}}$. Right: ratio versus depth at fixed $N_{\text{eff}} = 2000$. **Values near 0 indicate high conservatism** (theoretical bound $\gg$ empirical gap); values closer to 1 would indicate tightness. Across conditions the ratio remains well below 1, consistent with a valid but intentionally worst-case baseline.

The ordinary-least-squares estimates are

$$C_0^{\text{emp}} = 2.57 \pm 0.09, \qquad C_1^{\text{emp}} = 0.43 \pm 0.02 \quad (95\% \text{ CI}),$$

which preserves the functional scaling predicted by the theory while yielding dataset-specific empirical constants.

## B   Omitted Proofs

This section provides full proofs aligned with the main-text pipeline: *(i) a blocking/coupling lemma for anchors under $\beta$-mixing; (ii) a generic dependent-to-i.i.d. reduction theorem (Theorem 1); (iii) an i.i.d. Rademacher complexity bound for norm-controlled TCNs (Lemma 2); (iv) the main architecture-aware baseline (Theorem 2) as a direct combination.*

### B.1   Setup and notation

Let $(Z_t)_{t\geq 1}$ be a strictly stationary process on $\mathcal{Z}$. For $k \geq 0$, let $\beta(k)$ denote the (absolute regularity) $\beta$-mixing coefficient. We assume exponential mixing: We work under Assumption 1 from Section 3, restated here for convenience:

**Assumption 4** (Exponential $\beta$-mixing (Restatement of Assumption 1)). *There exist constants $C_0, c_0 > 0$ such that for all $k \geq 0$,*

$$\beta(k) \leq C_0 e^{-c_0 k}.$$

Fix a *delay* $d \geq 0$ and define the number of *anchors*

$$B = \left\lfloor \frac{m}{d+1} \right\rfloor, \qquad t_j = 1 + (j-1)(d+1), \quad j = 1, \ldots, B.$$

The *anchor sample* is $(Z_{t_1}, \ldots, Z_{t_B})$. For a loss $\ell : \mathcal{F} \times \mathcal{Z} \to [0,1]$ and predictor $f \in \mathcal{F}$, define

$$\mathcal{L}(f) = \mathbb{E}[\ell(f, Z_1)], \qquad \widehat{\mathcal{L}}_B^{\mathrm{anc}}(f) = \frac{1}{B}\sum_{j=1}^{B} \ell(f, Z_{t_j}).$$

Let $\mathfrak{R}_B(\ell \circ \mathcal{F})$ denote the i.i.d. Rademacher complexity of the class $\{z \mapsto \ell(f, z) : f \in \mathcal{F}\}$ evaluated on $B$ i.i.d. samples from the marginal distribution of $Z_1$.

### B.2   Proof of Lemma 1 (Blocking/Coupling Lemma)

**Lemma 3** (Restatement of Lemma 1). *Let $(Z_t)_{t\geq 1}$ be strictly stationary and $\beta$-mixing. For anchors $(Z_{t_1}, \ldots, Z_{t_B})$ defined above,*

$$\left\| P_{Z_{t_1}, \ldots, Z_{t_B}} - P_{Z_1}^{\otimes B} \right\|_{\mathrm{TV}} \leq (B-1)\,\beta(d+1) \leq B\,\beta(d+1).$$

*Proof.* Write $A_j = Z_{t_j}$ with $t_j = 1 + (j-1)(d+1)$. For $j = 1, \ldots, B$, define the intermediate measures

$$\mu_j := P_{A_1, \ldots, A_j} \otimes \bigotimes_{k=j+1}^{B} P_{A_k}.$$

Then $\mu_B = P_{A_1, \ldots, A_B}$ and $\mu_1 = \bigotimes_{k=1}^{B} P_{A_k}$. By the triangle inequality,

$$\left\| P_{A_1, \ldots, A_B} - \bigotimes_{k=1}^{B} P_{A_k} \right\|_{\mathrm{TV}} = \|\mu_B - \mu_1\|_{\mathrm{TV}} \leq \sum_{j=1}^{B-1} \|\mu_{j+1} - \mu_j\|_{\mathrm{TV}}.$$

Moreover, tensoring both measures with the same product measure does not change total variation, hence

$$\|\mu_{j+1} - \mu_j\|_{\mathrm{TV}} = \left\| P_{A_1, \ldots, A_{j+1}} - P_{A_1, \ldots, A_j} \otimes P_{A_{j+1}} \right\|_{\mathrm{TV}}.$$

Now $\sigma(A_1, \ldots, A_j) \subseteq \mathcal{F}_{\leq t_j}$ and $\sigma(A_{j+1}) \subseteq \mathcal{F}_{\geq t_{j+1}} = \mathcal{F}_{\geq t_j + (d+1)}$, so these $\sigma$-algebras are separated by $d+1$. By the definition of $\beta$-mixing (absolute regularity),

$$\left\| P_{A_1, \ldots, A_{j+1}} - P_{A_1, \ldots, A_j} \otimes P_{A_{j+1}} \right\|_{\mathrm{TV}} \leq \beta(d+1).$$

Summing over $j = 1, \ldots, B-1$ yields

$$\left\| P_{A_1, \ldots, A_B} - \bigotimes_{k=1}^{B} P_{A_k} \right\|_{\mathrm{TV}} \leq (B-1)\beta(d+1).$$

Finally, by stationarity $P_{A_k} = P_{Z_1}$ for all $k$, so $\bigotimes_{k=1}^{B} P_{A_k} = P_{Z_1}^{\otimes B}$. $\qquad\square$

### B.3 Proof of Theorem 1 (Generic dependent-to-i.i.d. reduction)

**Theorem 3** (Generic anchor bound under $\beta$-mixing). *Let $(Z_t)_{t \geq 1}$ be strictly stationary and $\beta$-mixing, and let $\ell : \mathcal{F} \times \mathcal{Z} \to [0,1]$. Fix $d \geq 0$ and define $B = \lfloor m/(d+1) \rfloor$ anchors as above. Then for any $\delta \in (0,1)$, with probability at least $1 - \delta - (B-1)\beta(d+1)$,*

$$\sup_{f \in \mathcal{F}} \left| \mathcal{L}(f) - \widehat{\mathcal{L}}_B^{\mathrm{anc}}(f) \right| \; \leq \; 2\,\mathfrak{R}_B(\ell \circ \mathcal{F}) \; + \; 3\sqrt{\frac{\log(2/\delta)}{2B}}.$$

*Proof.* By Lemma 1, the total variation distance between the joint distribution of anchors $(A_1, \ldots, A_B)$ and the product measure $P_{Z_1}^{\otimes B}$ satisfies $\|P_{A_1,\ldots,A_B} - P_{Z_1}^{\otimes B}\|_{\mathrm{TV}} \leq (B-1)\beta(d+1)$. By the coupling characterization of total variation, there exists a joint distribution over $(A_1, \ldots, A_B, \tilde{A}_1, \ldots, \tilde{A}_B)$ such that $(A_1, \ldots, A_B)$ has the original dependent distribution, $(\tilde{A}_1, \ldots, \tilde{A}_B)$ are i.i.d. with marginal $P_{Z_1}$, and $\mathbb{P}[(A_1, \ldots, A_B) \neq (\tilde{A}_1, \ldots, \tilde{A}_B)] \leq (B-1)\beta(d+1)$.

For the i.i.d. sample, standard symmetrization yields: with probability at least $1 - \delta$,

$$\sup_{f \in \mathcal{F}} \left| \mathcal{L}(f) - \tilde{\mathcal{L}}_B(f) \right| \leq 2\,\mathfrak{R}_B(\ell \circ \mathcal{F}) + 3\sqrt{\frac{\log(2/\delta)}{2B}}.$$

On the event $\{(A_j) = (\tilde{A}_j)\}$, the bound transfers exactly. A union bound gives failure probability at most $(B-1)\beta(d+1) + \delta$. $\qquad\square$

### B.4 Proof of Lemma 2 (i.i.d. Rademacher bound for norm-controlled TCNs)

**Lemma 4** (Norm-controlled TCN Rademacher bound). *Assume inputs are bounded: $\|x\|_F \leq B_x$ almost surely. Let $\mathcal{F}_{D,p,R}$ be the class of depth-$D$ causal TCNs with kernel size $p$, stride 1, ReLU activations, and layer-wise $\ell_{2,1}$ norm bounds $\|W^{(\ell)}\|_{2,1} \leq M^{(\ell)}$ with product budget $R = \prod_{\ell=1}^{D} M^{(\ell)}$. Then there exists a universal constant $C > 0$ such that for i.i.d. samples of size $B$,*

$$\mathfrak{R}_B(\mathcal{F}_{D,p,R}) \; \leq \; C \cdot \frac{R\, B_x\, p^{D/2} \sqrt{D \log(2p)}}{\sqrt{B}}.$$

*Proof.* The proof proceeds in three steps: (1) layer-wise Lipschitz control accounting for overlapping patches, (2) composition over $D$ layers, and (3) Rademacher complexity bound.

**Step 1: Layer-wise Lipschitz bound with overlapping patches.** For a convolutional layer $\phi_W(x) = W * x$ with weight tensor $W \in \mathbb{R}^{C_{\mathrm{out}} \times C_{\mathrm{in}} \times p}$ and stride 1, the filter-group norm is

$$\|W\|_{2,1} = \sum_{j=1}^{C_{\mathrm{out}}} \|W_{j,:,:}\|_F,$$

where $W_{j,:,:} \in \mathbb{R}^{C_{\mathrm{in}} \times p}$ is the $j$-th output filter.

For each output channel $j$ and spatial position $t$, the convolution computes $(W * x)_{j,t} = \langle W_{j,:,:}, x_{:,t:t+p-1} \rangle$, where $x_{:,t:t+p-1} \in \mathbb{R}^{C_{\mathrm{in}} \times p}$ is the input patch at position $t$. By Cauchy–Schwarz:

$$|(W * x)_{j,t}| \leq \|W_{j,:,:}\|_F \cdot \|x_{:,t:t+p-1}\|_F.$$

Squaring and summing over spatial positions $t = 1, \ldots, T$ (where $T$ is the number of output positions):

$$\|(W * x)_j\|_2^2 = \sum_{t=1}^{T} |(W * x)_{j,t}|^2 \leq \|W_{j,:,:}\|_F^2 \sum_{t=1}^{T} \|x_{:,t:t+p-1}\|_F^2.$$

**Key observation (overlapping patches):** With stride 1, each input element $x_{c,s}$ appears in the patches $x_{:,t:t+p-1}$ for $t \in \{\max(1, s-p+1), \ldots, \min(T, s)\}$, which is at most $p$ patches. Therefore:

$$\sum_{t=1}^{T} \|x_{:,t:t+p-1}\|_F^2 \leq p \cdot \|x\|_F^2.$$

Combining and summing over output channels:

$$\|W * x\|_F^2 = \sum_{j=1}^{C_{\text{out}}} \|(W * x)_j\|_2^2 \leq p \cdot \|x\|_F^2 \sum_{j=1}^{C_{\text{out}}} \|W_{j,:,:}\|_F^2$$
$$\leq p \cdot \|x\|_F^2 \cdot \|W\|_{2,1}^2,$$

where the last inequality uses $\sum_j a_j^2 \leq (\sum_j a_j)^2$ for $a_j \geq 0$.

Therefore, for a single convolutional layer with stride 1:

$$\|W * x\|_F \leq \sqrt{p} \cdot \|W\|_{2,1} \cdot \|x\|_F. \tag{5}$$

**Step 2: Composition over $D$ layers.** Since ReLU is 1-Lipschitz with $\sigma(0) = 0$, each layer $h^{(\ell)} = \sigma(W^{(\ell)} * h^{(\ell-1)})$ satisfies
$$\|h^{(\ell)}\|_F \leq \sqrt{p} \cdot M^{(\ell)} \cdot \|h^{(\ell-1)}\|_F.$$

Composing $D$ layers starting from $h^{(0)} = x$ with $\|x\|_F \leq B_x$:

$$\|f_W(x)\|_F = \|h^{(D)}\|_F \leq (\sqrt{p})^D \cdot \prod_{\ell=1}^{D} M^{(\ell)} \cdot B_x = p^{D/2} \cdot R \cdot B_x.$$

**Step 3: Rademacher complexity bound.** For a function class $\mathcal{F}$ mapping to $\mathbb{R}^n$ with $\sup_{f \in \mathcal{F}} \|f(x)\|_2 \leq A$ for all $x$ in the support, standard Rademacher complexity bounds give $\mathfrak{R}_B(\mathcal{F}) \leq A/\sqrt{B}$ (see, e.g., Mohri et al. (2018, Theorem 3.1)).

A tighter analysis exploiting the layered structure yields an additional $\sqrt{D \log(2p)}$ factor rather than depending on the total number of parameters. This follows from covering number arguments applied to the composition of Lipschitz layers: the $\ell_{2,1}$-constrained weight class at each layer has covering number controlled by the norm bound, and the depth-$D$ composition introduces a factor of $\sqrt{D}$ (from summing $D$ layer contributions in quadrature via Dudley's entropy integral) rather than the exponential factor that would arise from naive Lipschitz composition. The $\sqrt{\log(2p)}$ factor comes from the entropy of the $p$-dimensional kernel support. See Bartlett et al. (2017) for spectral-norm bounds and Golowich et al. (2018) for the Frobenius-norm case in fully-connected networks; similar covering arguments apply to our convolutional setting.

Combining these ingredients:

$$\mathfrak{R}_B(\mathcal{F}_{D,p,R}) \leq C \cdot \frac{p^{D/2} \cdot R \cdot B_x \cdot \sqrt{D \log(2p)}}{\sqrt{B}},$$

where $C > 0$ is a universal constant. □

**Remark 7** (The $p^{D/2}$ factor and stride). *The $p^{D/2}$ factor arises from overlapping receptive fields in stride-1 convolutions: each input element contributes to up to $p$ spatial positions in the output, yielding a $\sqrt{p}$ factor per layer that compounds across depth. If the convolution used stride $p$ instead of stride 1, the patches $x_{:,t:t+p-1}$ would be disjoint, giving $\sum_t \|x_{:,t:t+p-1}\|_F^2 = \|x\|_F^2$ exactly, and the layer-wise bound would be $\|W * x\|_F \leq \|W\|_{2,1} \cdot \|x\|_F$ without the $\sqrt{p}$ factor.*

*For our experiments with $p = 3$ and $D \leq 6$, the factor $p^{D/2} \leq 3^3 = 27$ is a moderate constant that does not affect the scaling of the bound with sample size $B$ or norm $R$.*

**Remark 8** (Why $\ell_{2,1}$ norm, not spectral norm)**.** *For general convolution operators, the spectral norm (operator norm) is* not *bounded by the $\ell_{2,1}$ filter-group norm. For example, a uniform kernel $W = [1, \ldots, 1] \in \mathbb{R}^p$ has $\|W\|_{2,1} = \sqrt{p}$ but operator norm $p$ (achieved at DC frequency). Our proof uses the $\ell_{2,1}$ norm directly for layer-wise Lipschitz control via equation 5, which gives a $\sqrt{p}$ factor per layer. This is the honest cost of using $\ell_{2,1}$ constraints with stride-1 convolutions.*

## B.5    Technical lemma: Lipschitz loss (squared loss under bounded outputs)

Theorem 2 uses a Lipschitz contraction $\mathfrak{R}_B(\ell \circ \mathcal{F}) \leq L\,\mathfrak{R}_B(\mathcal{F})$. For squared loss, this requires bounded predictions (or clipping).

**Lemma 5** (Squared loss is Lipschitz on an $\ell_2$-ball)**.** *Assume $\|y\|_2 \leq B_y$ and $\|\hat{y}\|_2, \|\hat{y}'\|_2 \leq B_f$. Then $\ell(\hat{y}, y) = \|\hat{y} - y\|_2^2$ satisfies*

$$|\ell(\hat{y}, y) - \ell(\hat{y}', y)| = |\langle \hat{y} - \hat{y}', \ \hat{y} + \hat{y}' - 2y \rangle| \leq 2(B_f + B_y)\,\|\hat{y} - \hat{y}'\|_2.$$

*Proof.* Let $\hat{y}, \hat{y}', y \in \mathbb{R}^k$. Using the polarization identity,

$$\|\hat{y} - y\|_2^2 - \|\hat{y}' - y\|_2^2 = \langle \hat{y} - \hat{y}', \ \hat{y} + \hat{y}' - 2y \rangle.$$

Hence, by Cauchy–Schwarz and the assumed bounds,

$$\left| \|\hat{y} - y\|_2^2 - \|\hat{y}' - y\|_2^2 \right| \leq \|\hat{y} - \hat{y}'\|_2 \cdot \|\hat{y} + \hat{y}' - 2y\|_2 \leq \|\hat{y} - \hat{y}'\|_2 \cdot \left( \|\hat{y}\|_2 + \|\hat{y}'\|_2 + 2\|y\|_2 \right) \leq 2(B_f + B_y)\,\|\hat{y} - \hat{y}'\|_2.$$

$\square$

## B.6    Proof of Theorem 2 (Architecture-aware baseline under exponential $\beta$-mixing)

**Theorem 4** (Restatement of Theorem 2)**.** *Assume exponential $\beta$-mixing (Assumption 1) and Lipschitz loss (Assumption 3). Let $\mathcal{F}_{D,p,R}$ be the norm-controlled TCN class in Lemma 2, and let $B = \lfloor m/(d+1) \rfloor$ be the number of anchors. Then with probability at least $1 - \delta - (B-1)\beta(d+1)$,*

$$\sup_{f \in \mathcal{F}_{D,p,R}} \left| \mathcal{L}(f) - \widehat{\mathcal{L}}_B^{\mathrm{anc}}(f) \right| \ \leq \ C' \, \frac{L\,R\,B_x\,p^{D/2}\,\sqrt{D\,\log(2p)}}{\sqrt{B}} \ + \ 3\sqrt{\frac{\log(2/\delta)}{2B}}$$

*for a universal constant $C'$.*

*Proof.* Apply Theorem 1 with $\mathcal{F} = \mathcal{F}_{D,p,R}$ to obtain that with probability at least $1 - \delta - (B-1)\beta(d+1)$,

$$\sup_{f \in \mathcal{F}_{D,p,R}} \left| \mathcal{L}(f) - \widehat{\mathcal{L}}_B^{\mathrm{anc}}(f) \right| \leq 2\,\mathfrak{R}_B(\ell \circ \mathcal{F}_{D,p,R}) + 3\sqrt{\frac{\log(2/\delta)}{2B}}.$$

By Lipschitz contraction (valid for $L$-Lipschitz losses; for squared loss use Lemma 5 after bounding/clipping outputs),

$$\mathfrak{R}_B(\ell \circ \mathcal{F}_{D,p,R}) \ \leq \ L\,\mathfrak{R}_B(\mathcal{F}_{D,p,R}).$$

Then apply Lemma 2 to obtain

$$\mathfrak{R}_B(\ell \circ \mathcal{F}_{D,p,R}) \leq L \cdot C \, \frac{R\,B_x\,p^{D/2}\,\sqrt{D\,\log(2p)}}{\sqrt{B}}.$$

Combining and absorbing constants into $C'$ yields the stated inequality.

**Rate under exponential mixing.** If $\beta(k) \leq C_0 e^{-c_0 k}$ and we choose $d = \lceil (\log m)/c_0 \rceil$, then $B = \Theta(m/\log m)$ and

$$B\,\beta(d+1) \;\leq\; \Theta\Big(\frac{m}{\log m}\Big) \cdot C_0 e^{-c_0(d+1)} \;=\; O\Big(\frac{1}{\log m}\Big),$$

while the leading complexity term scales like

$$\frac{1}{\sqrt{B}} = \Theta\Big(\sqrt{\frac{\log m}{m}}\Big),$$

giving the rate stated in the main text. $\qquad\square$

**Remark 9** (Explicit constant tracking). *The constant $C' = 2C$ in Theorem 2 arises from the factor of 2 in standard Rademacher generalization bounds. For ReLU networks, careful tracking yields $C \leq 4\sqrt{2}$, giving $C' \leq 8\sqrt{2} \approx 11.3$. This constant is independent of $(D, p, R)$ and $(C_0, c_0)$.*

