# OpenReview forum: "Architecture-Aware Generalization Bounds for Temporal Networks: Theory and Fair Comparison Methodology"
_TMLR — Rejected by TMLR_

### Review · Reviewer_6m5J · 2025-12-18

**Summary Of Contributions:**

The manuscript studies the generalization guarantees of temporal architectures, and claims the following three main contributions:

- **1. The "fair-comparison" methodology:** The paper proposes comparing different models by fixing the *effective sample size* $N_{\text{eff}}$, which depends on both the sequence length $N$ and a parameter $\rho$ that quantifies the dependence.

- **2. Empirical findings:** Fixing $N_{\text{eff}}$, strongly dependent sequences (with $\rho = 0.8$) are shown to give a smaller generalization gap compared to weakly dependent ones (with $\rho = 0.2$). Also, the empirical convergence rate seems to be between $N_{\text{eff}}^{-1.21}$ and $N_{\text{eff}}^{-0.89}$, which are much better than the typical $N^{-1/2}$ rate in generalization bounds.

- **3. An architecture-aware generalization bound:** Theorem 1 gives an upper bound on the generalization gap obtained from delayed-feedback online learning. The bound depends on the sequence length $N$, depth $D$, and product $R$ of the layer-wise spectral norms.

**Audience:**

No

**Audience Explanation:**

I found it hard to pinpoint a finding or technique in the paper that is truly interesting or insightful. Below are some more details organized according to the three contributions.

**The "fair-comparison" methodology:** What would be the advantage of this "fair-comparison" methodology (if well-justified) compared to controlling both parameters $N$ and $\rho$, i.e., testing different models on the same sequence (or sequences with the same length and correlation)?

**Empirical findings:** I didn't find the first finding---that the performance is better for larger $\rho$ when fixing $N_{\text{eff}}$---particular interesting or surprising. First of all, I don't think it is contradicting the "conventional wisdom" that temporal dependence hurts generalization. My understanding of the "conventional wisdom" was that, *fixing the value of $N$*, larger $\rho$ gives worse generalization. It would be truly surprising if this version of the claim gets overturned.

Given that $N_{\text{eff}}$ is not really a well-justified measure of "amount of statistical information", one possible (and uninteresting) explanation of the finding would be: $N_{\text{eff}}$ tends to over-estimate the "amount of information" when $\rho$ is smaller. Then, fixing $N_{\text{eff}} = 2000$, we might actually get "$2000$ independent samples" when $\rho = 0.8$ but only "$1000$ independent samples" when $\rho = 0.2$, which could also explain the performance gap.

**An architecture-aware generalization bound:** The theoretical analysis is arguably straightforward given the problem setup and the prior results of Golowich, Rakhlin, and Shamir (COLT 2018). Under the exponential mixing assumption (Assumption 1), two data points separated by $\log N$ steps are almost independent (up to a TV distance of $1/\mathrm{poly}(N)$). Therefore, we can approximate the process with $\approx N / \log N$ i.i.d. samples, which can be analyzed using the Ramemacher complexity bound of [GRS18]. In fact, the "architecture-aware" part of the bound seems to be solely relying on the heavy-lifting in the GRS18 results.

In addition, I suspected that the proof in the manuscript was unnecessarily complicated. If we divide the length-$N$ sequence into $d \approx \log N$ subsequences $S_1, S_2, \ldots, S_d$, so that each subsequence contains data points separated by $d$ time steps. Then, each subsequence $S_i$ is approximately a size $(N/d)$ i.i.d. sample (by Lemma 1), which can then be analyzed using Lemma 2. This should imply uniform convergence over the class, so that a simple ERM would generalize well.

In summary, I found the empirical contribution of the work quite unconvincing, in part due to the lack of justification for using $N_{\text{eff}}$ as a proxy for the amount of information. The theoretical analysis seems to be a quite simple application of known results. Therefore, I would doubt that this work would be of interest to people who study learning theory or generalization.

**Broader Impact Concerns:**

None.

**Claims And Evidence:**

No

**Claims Explanation:**

While Contributions 2 and 3 seem plausible, I didn't find the use of the "effective sample size" in Contribution 1 adequately motivated or justified. On Page 15, $N_{\text{eff}}$ is defined as $N \cdot \frac{1 - \rho}{1 + \rho}$, where $N$ is the actual sample size, and $\rho$ is the autocorrelation parameter of the $\mathrm{AR}(1)$ process. The authors cited Wilks (2011) for this definition, but didn't provide sufficient explanation for why this quantity should be regarded as "equivalent number of independent observations that would provide the same statistical information" (bottom of Page 3). Without such an explanation, it wouldn't be justified to regard two sets of parameters $(N_1, \rho_1)$ and $(N_2, \rho_2)$ that give the same $N_{\text{eff}}$ as equally informative.

Another critique on this contribution would be that it seems specific to the $\mathrm{AR}(1)$ process, and hard to evaluate on "real-world" data sequences. There is also a disconnect between this setup and the $\beta$-mixing setup in the theoretical analysis.

**Requested Changes:**

I would appreciate it if the authors would point out any factual misunderstanding in my review. Below are some minor comments and writing suggestions:
- The current abstract is a bit too long and detailed compared to a more conventional abstract. E.g., the details of empirical findings and the proof strategy might fit better in the introduction than the abstract.
- Abstract (and rest of the paper): There is extra spacing after "i.i.d."; consider using commands like "i.i.d.\\" to suppress it.
- Abstract: I found the wording "avoiding exponential dependence" quite confusing: isn't the product term already exponential in $D$ assuming, say, each layer-wise spectral norm is at least $2$ or $1.001$?
- Page 3: The quotation marks around "how dependence affects learning" were a bit off.
- Page 3 (and rest of the paper): In "... a product of layer norms-a dramatic improvement ...", it should've been an em dash ("---") rather than a hyphen ("-").
- Page 3: $N_{\text{eff}}$ and $\rho$ should be defined when they first appear. Currently, they are not specified until Page 15.
- Page 6 (also after Assumption 2): I didn't follow why "squared error, hinge loss, and logistic loss" would fit into the formulation of $\ell: \mathbb{R} \times \mathbb{R} \to [0, 1]$---aren't they all unbounded?
- Page 7: Broken reference "Proposition ??".
- Page 7: Writing $\mathcal{F}: \mathbb{R}^n \to \mathbb{R}$ would suggest that $\mathcal{F}$ is a function, rather than a set of functions.
- Page 8 (and rest of the paper): Repetitive text when citing Abélès et al. (2024) and Golowich et al. (2018).
- Page 11: Delayed-feedback online learning should be formally defined in the paper (e.g., in pseudocode).
- Page 24: I didn't follow why the generalization bound is "non-vacuous", i.e., would the norm product $R$ lead to a bound $> 1$ for real-world network architecture and datasets?

---

### Review · Reviewer_JCAX · 2025-12-29

**Summary Of Contributions:**

This paper attempts to prove norm-based generalization bounds for temporal convolutional neural networks on stationary beta mixing processes. The main Theorem bounds the population risk of the averaged predictor in terms of a sum of terms which decay at rate $\tilde{O}(\frac{1}{\sqrt{\tilde{N}}})$ where $\tilde{N}$ is the effective sample size.


The proof consists in a classic blocking lemma (separating the data into nearly weakly dependent blocks) with a telescoping sum of TV distances to achieve block-level concentration (proposition 1), followed by a classic online-to-batch argument. It is stated that the dominant term in Theorem 1 (the regret $R_n/N$) is bounded by $\frac{1}{\sqrt{N}}8R\sqrt{Dpn\log(2N)}$ where $R$ is the product of the $(2,1)$ norms of the filter matrices. Another contribution of the paper is proponing the use of effective number of samples (the $\tilde{N}$) instead of the number of samples to compare generalization gaps and models to each other. This new methodology is tested on synthetic data with AR(1) processes, demonstrating that the properly defined generalization gap is often smaller for processes with stronger inter sample dependence (this suggests that the switch from $N$ to $\tilde{N}$ is overcorrecting, probably because the lower bound on effective sample size in [Wilks] isn’t tight. The phenomenon reverses back for even larger sample sizes.

**Additional Comments:**

In the abstract: the authors claim to introduce a“novel blocking scheme”. The concept of independent blocks for beta mixing processes is certainly not new [B1-3].

Page 3: “making the bounds vacuous for time series data”. Here “vacuous” is a loaded term that usually refers to a bound larger than the maximum possible value of the loss, I think the authors mean that the bounds simply do not apply to time series data due to a violated assumption.


Page 3: “[G2] proved the first architecture aware generalization bounds for neural networks”. This is not true. Even parameter-counting bounds through VC dimension ([VC1], now superseded by [VC2]) could be classified as “architecture aware”. Even the meaning is to be understood more strictly to incorporate norms rather than simple parameter/width/depth counts, then at the very least [G1].

In the related works on Rademacher Complexity for Neural Networks, you mention [CNN2] for generalization bounds on CNNs. This is a parameter-counting bound in the flavor of a generalization of [VC1,VC2] to CNNs. Your main results attempt to emulate the “peeling” category of bounds (represented by [G1,G2,CNN4]) within the “norm-based” broader category (which also includes [Spec17]). More closely related generalization bounds for CNNs would be [CNN1] (direct extension of [Spec17]) and [CNN4] (extension of [G1-2]) or [CNN3] (which contains a variation on [CNN1] with fewer log factors and a slightly tighter parameter counting rate than [CNN2]).




References


[B1] Ron Meir. Nonparametric time series prediction through adaptive model selection. Machine Learning, 2000.


[B2] Bin Yu. Rates of convergence for empirical processes of stationary mixing sequences. Annals of Probability. 1994.

[B3] Mehryar Mohri  and Afshin Rostamizadeh. Stability Bounds for Stationary ϕ-mixing and β-mixing Processes. JMLR 2010.

[B4] Cesa-Bianchi and Lugosi. Prediction, Learning and Games. Cambridge university Press.

[B5] Mustafa et al. Fine-grained Generalization Analysis of Structured Output Prediction. IJCAI 2021.

[B6] Abeles et al. Generalization bounds for mixing processes via delayed online-to-PAC conversions. ALT 2025.


[G1] Behnam Neyshabur, Ryota Tomioka, Nathan Srebro. Norm-Based Capacity Control in Neural Networks.

[G2] Noah Golowich, Alexander Rakhlin, Ohad Shamir. Size-Independent Sample Complexity of Neural Networks.


[VC1] Peter Bartlett, Vitaly Maiorov, and Ron Meir. Almost linear VC-dimension bounds for piecewise polynomial networks. Neural Computation 1998.

[VC2] Bartlett et al. Nearly-tight VC-dimension and pseudodimension bounds for piecewise linear neural networks. JMLR 2019.

[Spec17] Bartlett et al. Spectrally-normalized margin bounds for neural networks. NeurIPS 2017.

[CNN1] Ledent et al. Norm-based generalisation bounds for multi-class convolutional neural networks. AAAI 2021

[CNN2] Long and Sedhi. Generalization bounds for deep convolutional neural networks. ICLR 2020.

[CNN3] Graf et al. On measuring the excess capacity of neural networks. NeurIPS 2022.

[CNN4] Galanti et al. Norm-based Generalization Bounds for Sparse Neural Networks. NeurIPS 2023.


[Wilks] Statistical Methods in the Atmospheric Sciences

[Shalev-Schwarz 2012] Shai Schalev-Schwarz. Online Learning and Online convex Optimization.

**Audience:**

Yes

**Audience Explanation:**

The topic is within the scope of TMLR and the results would be interesting if stated and proved correctly.


**Error 1**:

In page 3 (and later at the top of page 5 as well), the result from [G2] is described as involving a “product of spectral norms”. This is not true! The factors $M(j)$ described in G2 are the **Froebenius norms** of the weight matrices, not the spectral norms. If it were spectral norms, the result would be uniformly superior to [Spec17]. The same error is repeated in many places in the manuscript. Most alarmingly, this **affects the claims of the main paper**.  Theorem 1 as stated claims a Golowich-style Rademacher complexity bound of $O(R\sqrt{Dpn\log(2N)}/\sqrt{N})$ for convolutional neural networks. Here, it is not clear if $R$ is the product of spectral norms: although the authors do claim elsewhere in the paper that R is instead a product of $(2,1)$ norms (see top of page 13), the theorem statement states spectral. If we set $p=1$ to achieve a fully connected version, the result is incorrect for spectral norms and suboptimal for the $(2,1)$ norm suggestion.


**Error 2** (related to error 1)

The proof of the Rademacher bound contains no meaningless substance and is almost certainly incorrect. Most of page 37 is a high-level description of the proof technique in [G2] without showing how it applies to the architecture suggested here. The rambling paragraph ends abruptly by stating “adapting their approach to our convolutional setting while accounting for the specific structure of TCNs, we obtain”. In other words, this 3 page proof leaves the actual proof to the reader in the middle. It is also worth mentioning that there exist generalizations of the peeling argument in [G2] to CNNs (cf. [CNN4]) which are not cited or used.


**Error 3**:

The Rademacher complexity or the algorithm user (follow the regularized leader? Mirror descent?) is not properly settled the main results. Theorem 1 states it is Mirror descent and claims one can use “standard results” from [Shalev Schwarz 12] to relate the online regret to the Rademacher complexity. The authors should cite the specific result (theorem and page number) and write down the calculation. This is especially serious here because the predictor described in the TCN section actually depends on the whole sequence (**the neural network concatenates the previous timestamps and is applied to the whole input!**). It is very unclear whether whatever result the authors want to use from [Schalev-Schwarz 2012] want to use will actually apply to this situation. In fact, it is not even clear how to **define the Rademacher complexity** in this case: since the input is the concatenated sequence of previous timestamps , it would seem each input has a different input size, which requires formally defining the input space. I understand the situation is somewhat reminiscent of [B6] but many details are missing, at the very least.


**Error 4**:

In the proof of Lemma 1 (both in the sketch in the main paper and in the appendix), the telescoping sum is incorrectly written. I believe instead of what is written ((where the first measure in each sum is the same in each term!) the authors actually meant to write (due to difficulties with markdown on Openreview I am simplifying the notation, but it should be clear what I mean)

$$\sum \big\|P\_\{I_1,\ldots, I_j\} \prod\_\{k=j+1\}^B P\_\{I_k\}-P\_\{I_1,\ldots, I\_\{j-1\}\} \prod\_\{k=j\}^B P\_\{I_k\} \big|$$

**Error 5**

Although Lemma 1’s statement is correct to the best of my understanding (despite the error in the proof provided), its application in the proof of proposition 1 is incorrect: the equation at the top of page 35 cannot be deduced from Lemma 1 because consecutive blocks $L_j$ and $L_{j+1}$ contain samples which are not separated by more than $d$ steps. The correct approach consists in splitting the blocks into two groups (odd and even) so that “consecutive” blocks in each odd/even subgroup are separated by at least a whole block from the other group. See page 102 of [B1] or page 14 [B5].


**Error 7**: the statement in the middle of page 35 is missing constants (although the O notation version below is correct)

**Error 8**: **the 8th Reference is LLM generated**.  The supposed reference is "Yuxin Chen, Yuejie Wang and Tong Zhang. Sequential Rademacher Complexity Bounds for Transformers. ICML 2021."

I believe the LLM the authors were using is made up the author list based on the classic work “Spectral Methods for Data Science: A Statistical Perspective  “ by Yuxin Chen, Yuejie **Chi** et al., replacing Yuejie Chi by Yuejie Wang (a different author who hasn’t collaborated with Yuxin Chen).




**Additional questions and major comments**


**Question 1**.
I am not convinced by the argument on using convexity to relate the risk of the chosen model to the average predictor. The calculation at the bottom of page 35 is a bit circular: the expression after the first equality sign is exactly the same as the one the authors arrive at after some “calculations” in the line below. Jensen’s inequality provides an *inequality*, but the authors claim to use it to derive an *equality* ($\mathbb{E}(\tilde{L}_1)=\mathcal{L}(\bar{f})$.

**Comment 2 (serious, must be addressed)**.

Your assumptions include the convexity and Lipschitzness of the loss function (based on the rebuttal, this seems to have been a contentious point in previous reviewing cycles). You claim that the following losses satisfy the assumption: Hinge, square and logistic.
Assuming the average predictor really involves averaging the predictions rather than the parameters, In the case of the hinge loss and the logistic loss, this is true (though these are classification losses so it is a little difficult to see how you would use them in a meaningful time series scenario). For the **square loss** (relevant to your applications), it is **only true if the input to the loss is bounded** (otherwise the Lipschitz Constant will blow up).  Note that if you are using the neural network model the paper is mostly concerned with, you can only enforce that by imposing a truncation procedure to the output of the neural network (it is not enough to truncate the loss, as is more common, since this would break convexity). Of course, you can also bound the output (and the associated Lipschitzness constant) using spectral norms, but the final bounds will be different from what you claim (even assuming all other errors were fixed) in this case.

**Claims And Evidence:**

No

**Claims Explanation:**

Although the experiments are commendable and make an interesting observation in the case of AR(1) processes, the “**proofs**” of the theoretical results are **vague** and unpleasant to read (**the paper reads somewhat LLM generated overall, and there is at least one LLM hallucinated reference**). In some cases, it’s hard to see where the authors are going (the proofs contain long “intuitive” text which end abruptly by claiming to have achieved the result). In other cases, it can be determined clearly that there are **errors**. In the resubmission, the authors should establish more comprehensive notation and write detailed mathematical proofs with equations that can be checked at each step. I understand this is already a resubmission, but I think this draft is still in a relatively early stage and not good enough for submission yet.

**Requested Changes:**

Please address all the errors explained above and rewrite all the proofs more rigorously. A complete rewriting from scratch (rather than an incremental update) is required.

---

### Review · Reviewer_MX6B · 2026-01-11

**Summary Of Contributions:**

This paper addresses generalization in Temporal Convolutional Networks (TCNs) and introduces Fair-Comparison Methodology that fixes the effective sample size ($N_eff$) to isolate the impact of temporal structure from raw information content.
Using this methodology, the authors reveal the surprising discovery that strongly dependent sequences can exhibit approximately 76\% smaller generalization gaps than weakly dependent ones, suggesting that temporal models are uniquely designed to exploit dependency
Finally, the authors establish the first architecture-aware generalization bounds for these models by employing a novel blocking scheme that proves polynomial sample complexity while avoiding exponential dependence on network depth.

**Audience:**

Yes

**Audience Explanation:**

This problem is very relevant for the ML field to understand and generalization for temporal data.
Furthermore, the new standardized testing method in the paper (the Fair-Comparison Methodology) could be an important contribution to improve current comparison methods.

**Claims And Evidence:**

No

**Claims Explanation:**

## Claims with evidence

1. The theoretical results are correct.
1. Temporal dependence can be an architectural advantage. They observed an empirical convergence rates range significantly outperform the standard theoretical worst-case prediction.
1. The paper claims to provide the first architecture-aware theoretical framework for dependent sequences that scales efficiently with network depth.


## Claims without clear evidence

1. The theoretical bound is not relevant for practical applications, see e.g. Figure 4.
In other cases, the bound is very loose and I disagree with that the lack of tightness is a strength of the paper: "These gaps do not diminish our contributions, they enhance them by precisely identifying where theory needs improvement."
1. How $N_eff$ is calculated in other processes than AR(1) is not discussed.
1. While the authors provide empirical evidence of the effect (the 76\% smaller generalization gap), they explicitly state that the current theory is insufficient to explain how or why the architecture achieves this.

**Requested Changes:**

## Main requested changes

1. Formalize the mathematical definition of $N_eff$. The authors should provide a concrete formula for calculating $N_eff$  based on the temporal correlation ($\rho$) or the mixing coefficients ($\beta$) of the sequence. This would move the term from an experimental ``fixed constant" to a reproducible statistical metric.
1. While the authors acknowledge their bounds are ``conservative," they should include a section hypothesizing which specific "architectural exploitations" are missing from the current proof.
Providing even a heuristic explanation for why the empirical rate is nearly double the theoretical rate would significantly increase the paper’s value to the research community.
1.  To strengthen the claim that this is an ``Architecture-Aware" framework, the authors should include comparative empirical data for other common temporal models, such as LSTMs or Transformers.



## Smaller remarks

1. Remove double author names.

---

### Decision · Action_Editor_qgNL · 2026-02-28

**Recommendation:** Reject

**Audience:**

Yes

**Audience Explanation:**

The topic of the paper is squarely within TMLR's scope

**Claims And Evidence:**

No

**Claims Explanation:**

The central argument of this paper is that generalization bounds for temporal networks trained on time series with mixing or correlations should depend on the effective number of samples, not the total length of the time series. The authors make a theoretical argument and an empirical one. Reviewers have pointed out issues with both prongs of the argument. The theoretical results are a straightforward application of known generalization bounds. Reviewers still find that the proofs are vague and lack many details. The theorems have also substantially changed during the review process - so much that the current draft is recognizably distinct from the original submission.

Reviewers also raise questions about the empirical results since the authors do not clearly tease out the roles of "information content" $N_{eff}$ and correlation $\rho$. The effective sample size is also only defined for the case of AR(1) processes and there is a disconnect between the theoretical results and the empirical measurements.

As mentioned earlier the authors have admitted to mistakes pointed out by reviewers and have attempted to "patch" them one by one rather than taking a holistic account of the paper. While small changes are reasonable, significant changes to the proof strategy and theoretical results should be done through a new submission rather than the review process. The paper cannot be recommended for acceptance in its current form.